# Dynamics of aerosol, humidity, and clouds in air masses travelling over Fennoscandian boreal forests

Meri Räty[1], Larisa Sogacheva[2], Helmi-Marja Keskinen[1*], Veli-Matti Kerminen[1], Tuomo Nieminen[1,3], Tuukka Petäjä[1,4], Ekaterina Ezhova[1], Markku Kulmala[1,4,5]

[1]Institute for Atmospheric and Earth System Research (INAR) / Physics, Faculty of Science, University of Helsinki, Helsinki, Finland
[2]Climate Change Programme, Finnish Meteorological Institute, Helsinki, Finland
[3]Institute for Atmospheric and Earth System Research (INAR) / Forest Sciences, Faculty of Agriculture and Forestry, University of Helsinki, Helsinki, Finland
[4]Joint International Research Laboratory of Atmospheric and Earth System Sciences, School of Atmospheric Sciences, Nanjing University, Nanjing, China
[5]Aerosol and Haze Laboratory, Beijing Advanced Innovation Center for Soft Matter Sciences and Engineering, Beijing University of Chemical Technology (BUCT), Beijing, China
*Current: Independent researcher, Finland

*Correspondence to:* Meri Räty (meri.raty@helsinki.fi)

**Abstract.** Boreal forests cover vast areas of land in the high latitudes of the northern hemisphere, which are under amplified climate warming. The interaction between the forests and the atmosphere are known to generate a complex set of feedback processes. One feedback process, potentially producing a cooling effect, is associated with an increased reflectance of clouds due to aerosol-cloud interactions. Here, we investigate the effect that the boreal forest environment can have on cloud-related properties during the growing season. The site investigated was the SMEAR II station in Hyytiälä, Finland. Air mass back trajectories were the basis of the analysis and were used to estimate the time each air mass had spent over land prior to its arrival at the station. This enabled tracking the changes occurring in originally marine air masses as they travelled across the forested land. Only air masses arriving from the north-western sector were investigated, as these areas have a relatively uniform forest cover and relatively little anthropogenic interference. We connected the air mass analysis with comprehensive in-situ and remote-sensing data sets covering up to eleven growing seasons. We found that the properties of air masses with short land transport times, thereby less influenced by the forest, differed from those exposed to the forest environment for a longer period. The fraction of air masses with cloud condensation nuclei concentrations (at 0.2% supersaturation) above the median value of 180 cm-3 of the analysed air masses, increased from approximately 10% to 80% by 55 h of exposure to boreal forest, while the fraction of air masses with specific humidity above the median value of 5 g/kg increased from roughly 25% to 65%. Signs of possible resulting changes in the cloud layer were also observed from satellite measurements. Lastly, precipitation frequency increased from the average of approximately 7% to about 12% after a threshold of 50 hours of land transport. Most of the variables showed an increase with an increasing land transport time until approximately 50-55 hours, after which a

balance with little further variation seemed to have been reached. This appears to be the approximate time scale in which the forest-cloud interactions take effect, and the air masses adjust to the local forest environment.

## 1 Introduction

Boreal forest is a distinct biome dominated by evergreen trees that extends over an area of approximately 15 million square kilometres throughout the northern mid and high latitudes, representing over a third of Earth's total forest area (Bonan, 2008). As with any biome, the prevailing climate, such as the large seasonal temperature variations, controls the extent and conditions of the forest. Conversely, the boreal forest also influences the climate on both local and global scales through the exchange of energy and carbon, but also via contributing to the hydrological cycle and by acting as a source of atmospheric aerosol (e.g. Bonan, 2008; Spracklen et al., 2008; Kulmala et al., 2004, Scott et al., 2014).

During the growing season, biogenic secondary organic aerosol (SOA) dominates the aerosol population in the atmosphere over the boreal forest (Heikkinen et al., 2020). In this photosynthetically active period, the forest can maintain a 1000-2000 $cm^{-3}$ loading of aerosol particles in a size range of ~40-100 nm (Tunved et al., 2006), in contrast to for example loadings of only some hundreds $cm^{-3}$ over marine areas (Heintzenberg et al., 2000; Zheng et al., 2018). Most of the mass of these particles is organic, consisting of condensed low volatility vapours that have been formed through oxidation of biogenic volatile organic compounds (BVOCs) emitted by the vegetation (Heikkinen et al., 2020). Globally, more than half of BVOCs is estimated to be isoprene ($C_5H_8$) (Guenther et al., 2012; Sindelarova et al., 2014), but in European boreal forests, the majority BVOC group emitted by the local vegetation is monoterpenes ($C_{10}H_{16}$), while a small contribution, in addition to isoprene, is coming from sesquiterpenes ($C_{15}H_{24}$) (e.g., Wang et al., 2017; Hellén et al., 2018; with Tarvainen et al., (2007) estimating their relative emission fluxes to be approx. 84%, 9% and 7%, in Finnish forests, respectively). In comparison to isoprene, monoterpene oxidation is comparatively potent at producing low volatility vapours (Ehn et al., 2014; Jokinen et al., 2015), and for example in a chamber photooxidation experiment by Lee et al. (2006), different monoterpene compounds led to SOA yield percentages that were 12-29 times higher than for isoprene. Consequently, monoterpenes have been considered the main contributor to SOA mass over boreal forests during summer (Tunved et al., 2006). BVOC concentrations have a high seasonal variability and are primarily emitted in the growing season as their emissions are largely tied to the photosynthetic activity of plants (Rantala et al., 2015). Their emissions rates also tend to show exponential correlation with temperature (e.g., Hellén et al., 2018; Tingey et al., 1980; Lappalainen et al., 2009; Filella et al., 2007) and can also vary in response to other environmental stressors (e.g., Peñuelas and Staudt, 2010; Loreto and Schnitzler, 2010; Taipale et al., 2021).

Significant amounts of new SOA can form rapidly from available suitable vapours in new particle formation (NPF) events (Mäkelä et al., 1997; Dal Maso et al., 2005). Days having a well-defined NPF event vary in frequency over the boreal region, with examples ranging from only 1.5 % of days in a 3-year campaign at a remote continental Siberian site (Wiedensohler et al., 2019), to 23% of days in a year on average in the Finnish boreal forest (Nieminen et al., 2014). Although there are exceptions (Zhang et al., 2021; Kulmala et al., 2017, 2021), high condensation sinks (CS) can often inhibit NPF, especially in

clean sites (e.g. Birmili et al., 2003; Hyvönen et al., 2005) where the more pristine air masses are typically more favourable for NPF (Sogacheva et al., 2005, Dada et al., 2017).

Atmospheric aerosol particles can absorb or scatter radiation, and consequently influence the radiation balance at the surface (Scott et al., 2014). They however also have a significant indirect effect on climate, through aerosol-cloud interactions: Every cloud droplet in the atmosphere is condensed around an aerosol particle that may vary in size depending on humidity and the particle's properties, but typically has a dry diameter of 50-100 nm or larger (Paasonen et al., 2018; Pruppacher and Klett, 2010; Kerminen et al., 2012). These particles that can activate as a nucleus for a cloud droplet, are called cloud condensation

nuclei (CCN). The number of available CCN affects the number and size of droplets in clouds, and a higher CCN concentration for example, means that the available water can distribute to a larger number of smaller droplets, leading to a more reflective cloud and an extended cloud lifetime (Twomey, 1977; Christensen et al., 2020). Because the boreal forest produces organic vapours that partake in the formation and condensational growth of aerosol particles, Spracklen et al. (2008) estimated that boreal forests could potentially be doubling the regional CCN concentrations (from 100 to 200cm$^{-3}$), making a difference in

the cloud radiative forcing between -1.8 and 6.7 Wm$^{-2}$.

In addition to providing the nuclei for cloud droplets, the boreal forest also provides the condensing water vapour. While some of the life-sustaining forest rainfall flows back into the oceans through rivers and lakes, or is filtered into groundwater, a significant portion is evapotranspired: either evaporated from wet surfaces or transpired by vegetation back into the atmosphere. For example, in a forest site in Sweden, total evapotranspiration during a growing season was estimated to be

85% of precipitation, 75% of it being transpired and evaporated from the tree canopy (Kozii et al., 2020). Therefore, for example forest harvesting typically decreases evapotranspiration and increases runoff (Wei et al., 2022). A model sensitivity analysis by Wei et al. (2018) suggested that the runoff in the boreal forest could be more sensitive to a 5% change in vegetation cover than the average across different forest biomes. Moreover, besides forests' effect on CCN and water vapour, forest surface properties also affect the mechanisms of distributing moisture into the atmosphere. In comparison to open lands, forests

absorb more radiation and have a high surface roughness, generating a thicker atmospheric boundary layer, which can promote effective uplift for the moisture provided by the surface evapotranspiration (Bosman et al., 2019; Teuling et al., 2017, Xu et al., 2022).

The water vapour expelled by the forest into the atmosphere, can be "recycled" and go on to form new clouds and precipitation. Globally, total continental evapotranspired water vapour is a significant source of precipitation, and especially remote

continental areas are highly dependent on recycled continental water (van der Ent et al., 2010). Some estimates of the average contribution of the total recycled terrestrial water vapour to the precipitation falling over continents range from 40% (van der Ent et al., 2010) to around 60 % (Schneider et al., 2017), but for example in China, rainfall can be up to 80% recycled from the water vapour air flows have picked up while crossing the Eurasian continent (van der Ent et al., 2010).

Together, we can expect the CCN and atmospheric moisture provided by the boreal forest to boost the cloud cover in the

growing season. Previous satellite-based analysis by Duveiller et al. (2021) indeed suggested that boreal forest cover can promote ca 5 % higher levels of cloud fractional cover in comparison to unforested land in summer and autumn.

Correspondingly, Xu et al. (2022) observed a local cloud reduction (-0.2%) associated with forest loss in East Siberia (2002-2018), which they attributed to resulted changes in moist convection.

The outlined interactions between the boreal forest and the atmosphere are expected to change in the future under the warming climate. BVOC concentrations for example, can change in response to vegetation shifts, changes in photosynthesising biomass, and in response to rising temperatures, $CO_2$ levels, and due to other environmental stressors; overall, being considered likely to increase (Peñuelas and Staudt, 2010). There is however significant uncertainty in the response of BVOCs to climate change as a whole, due to complex interactive and sometimes opposing effects, and varied response depending on the BVOC species (Feng et al., 2019; Carslaw et al., 2010). However, in response to raising temperatures some modelling studies have for example suggested monoterpene emissions to increase significantly (e.g., by 58% with 4.8 K temperature increase (Liao et al., 2006), and by 29% in the lowest 5 km of the atmosphere in latitudes above 50°N with a 6 K temperature increase (Boy et al., 2022)). In the boreal zone, increasing BVOCs have been proposed to possibly lead to a negative feedback effect (Kulmala et al., 2004), as the subsequently growing availability of condensable vapours could lead to higher SOA and CCN number concentrations (Paasonen et al., 2013). Sporre et al., (2019) for example, modelled that, globally, doubling of $CO_2$ levels with corresponding temperature increase could lead to an enhanced negative cloud forcing of -0.43$Wm^{-2}$, with the feedback being strongest in downwind tropical and boreal forest. Locally, in the Finnish boreal forest, Yli-Juuti et al. (2021) estimated a combined summertime radiative feedback of -0.63$Wm^{-2}°C^{-1}$ from cloud albedo and direct aerosol effects in response to increasing biogenic SOA. Due to the complexity of aerosol-cloud interactions, all estimates on future effect still have large uncertainties. Many studies connecting forest-cloud interactions are based on modelling efforts or satellite analysis, but long-term atmospheric observations can also provide a reliable way to assess the effect of forests on aerosol and water vapour, and ultimately clouds and precipitation.

Several earlier publications have investigated the interaction between the boreal forest and the atmosphere, by utilising air mass transport history. In the Finnish boreal forest, Tunved et al. (2006) for example, showed a correlation between aerosol mass and time of transport over forest-covered land, linking it to the accumulation of terpenes in these air masses; findings, that were later confirmed by Liao et al., (2014) who also observed an associated increase in the average particle size. Similar observations have also been made by Asmi et al. (2016) in an arctic Russian site. Petäjä et al. (2022) significantly expanded on this analysis by connecting aerosol and cloud observations. In addition to confirming previous findings, they observed the link between total particle number concentration and NPF frequencies that went from extremely common to much rarer with increasing transport time over the forest. They also saw an increase of CCN concentrations by at least a factor of 4, between 20 and 75 hours of over-land transport, and showed a consequent doubling of cloud droplet number concentrations, with a simultaneous increase in cloud liquid water paths between the same air masses. Their findings were limited to investigating a period of eight months.

In this paper, we aim to similarly capture large-scale interactions between the boreal forest environment and clouds, by investigating a comprehensive data set from 11 years. The starting point is in characterising air mass transport history and connecting land transport times of clean, initially marine air masses, with changes occurring in the concentrations of different

sized aerosol particles, and the humidity, as they travel in the forest environment. We also investigate how these changes might translate to changes in cloud optical properties and ultimately the amount of precipitation falling back to the surface. We consider air masses with very short transport times over land to be representative of relatively marine air masses, whereas air masses spent longer periods of time over the forest-covered land are expected to show clearer signs of having been influenced by the forest. With this methodology, and through the comparison of the closer-to-marine and the increasingly more terrestrial air masses, we can examine the interactions outlined here, and estimate the temporal scale in which the air masses transition and reach a continental steady state with the sources and sinks of water vapour and aerosol in the forest environment.

## 2 Methods

Our analysis was based on hourly air mass back trajectories, investigated with coinciding in-situ measurements and remotely sensed variables. The site of the measurements was the SMEAR II (Figure 1) station in Hyytiälä (61°51´N, 24°17´E, 170 m asl), located in the Finnish boreal forest (Hari and Kulmala, 2005). The population in the area is sparse and the nearest urban area of significant size is the city of Tampere (pop. ~250 000), located approximately 50 km southwest from the site.

We concentrated on observations from an 11-year-long period, 2006-2016, except for CCN data, for which we had measurements from eight years (2009-2016). Only the growing season (April-September) was analysed, as we were specifically interested in investigating the biogenic influences of the forests, which are important only when the vegetation is biologically active (e.g. Aalto et al., 2015; Hari et al., 2017). All the datasets used in our analysis are listed in Table 1.

The hourly air mass back trajectories used in the analysis had been modelled with the Hybrid Single-Particle Lagrangian Integrated Trajectory model (HYSPLIT) (Stein et al., 2015; Draxler and Hess, 1998). The trajectories had been routinely produced throughout the years, and the meteorological input had been updated whenever a new input was deemed more suitable for the purpose, or the old data were no longer produced. Therefore, the trajectories utilised had been modelled with several different NCEP meteorological model archive data, FNL for 2006, GDAS one-degree data for 2007-2013 and GDAS half-degree data for 2014-2016 (Table 1). The last one had a horizontal resolution of 0.5°, while in the other two it was 1°. The output frequency of these meteorological model data is 3 hours. The arrival height of the trajectories was 100 metres. Each trajectory covered a 96-hour long air mass location history with a 1-hour temporal resolution.

We focused our analysis only on the air masses arriving from a sector between the west and north. This is the typical sector to focus on when wishing to specifically examine clean air masses arriving at the site (e.g. Tunved et al., 2006; Petäjä et al., 2022). More specifically, we selected trajectories that were 90% or more within the area indicated in Figure 1. As there are no large cities or other substantial pollution sources in this area, we can expect most surface effects on the air mass to be biogenic. Anthropogenic influences can, however, not be fully ruled out as, while relatively sparsely populated, the area does include towns, other built environments and agricultural land.

For the selected "clean sector" trajectories, we calculated the 'Time over Land' (ToL) value. It included the time spent over continental land mass in our chosen area, while any time spent over islands (e.g., Greenland and Iceland) was ignored. Therefore, in practice, ToL refers to time spent over Fennoscandia in this context. The ToL was not necessarily a continuous

period on land, as some air masses spent some time over the Baltic Sea in between. Ideally, for ToL to reflect the time of interaction of the trajectories with the land below, they should not be travelling long times very high above the well mixed boundary layer, where they might be decoupled with the surface. The mean fraction of total trajectory points on land that were also below 500 metres was 83% (median 91%), and below 1000 metres, 95% (median 100%). Therefore, the trajectories were most of the time below typical daytime boundary layer heights (Sinclair et al., 2022) while over land. (And even if momentarily

above thin nocturnal boundary layers, still within the residual layer than can still be influenced by the forest environment (Lampilahti et al., 2021).)

The HYSPLIT single-line back trajectories are simplified air mass travel paths that have uncertainties. Estimations suggest that the absolute trajectory error can often be between 15-30% of the travel distance (Draxler and Rolph, 2007). These uncertainties naturally carry over to our air mass classification (by sector) and ToL values. However, as we are not focusing

on individual trajectory paths, but only on the general source region and an approximate characterisation of the path in relation to a relatively uniform large area of land in the form of ToL, the accuracy should be sufficient for the purpose. In addition, by investigating a very long and extensive dataset, we expect a reasonable ratio between reliable signal and inaccuracies in individual trajectories.

Roughly 75% of the land area of Finland and Sweden, as well as above 45% of the land area of Norway are covered by forests

or other wooded land (European Commission, 2021). A distribution of the forests (Kempeneers et al., 2011; Päivinen et al., 2001; Schuck et al., 2002) is shown visually in the map of Figure 1. It shows the high forest coverage that dominates in the region, with the only exceptions being the northernmost tundra and the western mountain range, which lack tree cover, but are also vegetated and pristine, and comparatively narrow tracts of land. Considering the high forest coverage in the land area investigated, we can consider ToL an approximate proxy for describing the time an air mass is influenced by the forest

environment, especially within the range of the other already discussed uncertainties.

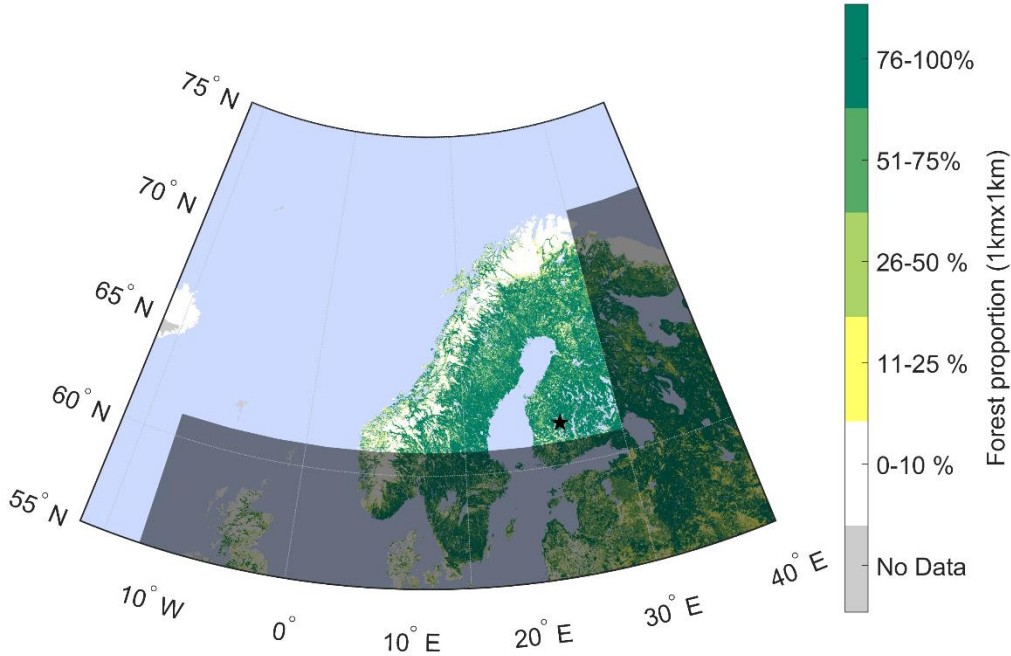

Figure 1. Map indicating the location of the SMEAR II station (black star) and the "clean sector" (not shaded). Only trajectories spending 90% or more within the clean sector were included in the analysis. Shown is also the proportion of forest cover in 1kmx1km resolution, based on the Forest Map of Europe produced by European Forest Institute (EFI) for the year 2011 (Kempeneers et al., 2011; Päivinen et al., 2001; Schuck et al., 2002).

To investigate differences in air masses with different ToLs, routinely measured meteorological datasets from Hyytiälä (temperature (T), relative humidity (RH), pressure ($p_0$), dew point temperature $T_d$, and precipitation (P)) were used. Specific humidity (q) was the only parameter not directly measured but it was derived from measured $p_0$ and $T_d$ (following formulations provided by WMO, 2018). Original minute-resolution data were converted to 1-hour resolution by taking hourly medians. Precipitation was only considered in terms of accumulation within the next 1 to 3 hours after the arrival of an air mass, which was a sum of initial 1-minute accumulation data. When a multi-hour rainfall was investigated, only periods in which the source region remained within the clean sector were accepted.

The CCN data used in this analysis was measured with a Cloud Condensation Nuclei Counter (CCNC; model CCN-100, Droplet Measurement Technologies). It was run in parallel with a Condensation Particle Counter (CPC, model TSI 3772) measuring the total particle number concentration in the sampled air ($N_{CN}$). The CCNC instrument consists of a saturation unit and an optical particle counter, which determines the number concentration of aerosol particles activated as CCN. A more detailed descriptions of the instrument and measurements can be found in literature (Paramonov et al., 2013; Paramonov et al.,

2015; Roberts and Nenes, 2005; Rose et al., 2008; Schmale et al., 2017). Different sized particles can be activated into CCN by varying the supersaturation ($S_{eff}$) in the CCNC chamber. In this analysis, we examine measurements conducted at supersaturations of 0.2 and 0.5 %. The measurements were not uniformly distributed between the years, as the supersaturation
settings and measurement frequency varied between the years (see availabilities in Table 1). For another source of aerosol data, we also utilised near-ground measurements of aerosol number size distribution made with the Differential Mobility Particle Sizer (DMPS; Aalto et al. 2001).

Remotely sensed parameters, cloud optical thickness (COT), cloud fraction (CF), and cloud water path (CWP), were acquired from the MODIS Level-2 Cloud Product (Platnick et al., 2017), based on the retrievals by the satellites MODIS Terra and
220 Aqua. Only the main COT and CWP datasets were used (variables *Cloud_Optical_Thickness* and *Cloud_Water_Path*), which include observations made only from 1km×1km pixels the cloud mask has classified as likely to be entirely cloudy (i.e. overcast) (Platnick et al., 2018). All COT and CWP observations that were within the valid range (i.e., not fill values) were included and no additional weighing, selection or filtering for the pixels was implemented. Therefore, pixels could for example include any cloud phases, and clouds at different altitudes, although we can expect most of the clouds to have been low-level
liquid clouds (Ylivinkka et al., 2020). The 5km×5km-resolution CF product (*Cloud_Fraction*), which is based on the fraction of 1km-resolution subpixels flagged with a cloud mask (of "probably cloudy" or "confident cloudy") (Platnick et al., 2018), was similarly used without additional filtering. We took medians and means from the satellite pixels from an area of 1 longitudinal and 0.5 latitudinal degrees, with the SMEAR II station in the centre. At this latitude, this area corresponds to approximately 52-53 km and 55.6 km, in zonal and meridional distance, respectively. The two satellites are in polar orbit, and
at 8-12 UTC (10-14 EET) they take together typically 4 (occasionally 5) images at least partially covering our area of interest. Some satellite views over the area are partial and cloud optical properties are only retrieved for cloudy pixels. This means that at worst an image could only have a few observation pixels. Therefore, we defined a threshold, in which the averages calculated, had to be based on a number of pixels at least 20% of the observed maximum pixel coverage. For CF, this in practice means that the satellite image had to cover at least 20% of the $1° \times 0.5°$ grid, whereas for COT and CWP which are
only derived for overcast cloudy pixels, this could be either a similar partial, but fully cloudy image, or even a full image of the grid but with overcast cloud pixel coverage being only 20% of the observed maximum. Files were processed file-by-file, and partial views resulting from subsequent files cutting over our area of interest (0.7% of all observations) were not combined into a single image. A few sporadic observations outside the main daytime window (3.7% of all images) were also discarded both for consistency and because we expect them to be less reliable for measurements based on shortwave radiation. To match
the satellite averages with the hourly trajectories, we applied rounding to the nearest hour, and in case of several satellite images within the same hour, took a pixel-weighed average. Despite starting with 4-5 daily images that averaged into 2-4 datapoints per day, we ended up with a much smaller satellite dataset after restricting our investigation only to images with sufficient spatial coverage, and further selecting only the cases coinciding with a clean-sector trajectory.

In most of our analysis, we divided observations into four value ranges to compare the variation in the fraction of observations
belonging to each range in different ToL-bins. Unless otherwise stated, the value ranges are based on quartiles, so that each

group contains a quarter of all observations. The variance in these complex natural variables is significant, but with this approach, it is easy to observe the changes in the fraction of cases belonging to the higher or lower value ranges. Because in many of our figures the data is divided into 5-hour ToL bins, we use 92.5 hours as a cut-off limit for ToL. The next bin would not be a full 5-hour bin and in the event of full trajectories (96 h) being located over land, accurate determination of total ToL is not possible as it can then be longer than the trajectory length. For consistency, this same cut-off is also used when data is binned hourly. Bins at the shortest ToLs that had only 5 or less data points were also omitted.

In support of visual observations, we utilised two tests to locate change-points, where a variable ceased being dependent of ToL. Firstly, we used Pettit's test on the bin median values (or means, when analysing precipitation), which can locate a statistically significant change-point in the central tendency. Secondly, we fitted least-square regression lines to the data points, starting from each ToL bin edge until the end, until we identified the first bin where the regression line became flat. This would clearly indicate no further least-square linear relationship between the two data sets.

**Table 1. A summary of the datasets used in this study. The meteorological inputs for the HYSPLIT simulations were from the U.S. National Oceanic and Atmospheric Administration's (NOAA) Air Resources Laboratory's (ARL) archives of the National Centers for Environmental Prediction (NCEP) meteorological model data. This included data from the FNL archive, and two Global Data Assimilation System (GDAS) archives (freely available on their website). Satellite observations were averaged over a 1°x0.5° geographical grid, which at this location corresponds to approximately 53 km x 56 km. The numbers in brackets in "Availability" indicate variation between years.**

| Dataset | Temporal coverage (Apr-Sep) | Availability (%) in clean sector air masses | Hourly resolution achieved | Additional | Reference |
|---|---|---|---|---|---|
| **HYSPLIT 96-hour back trajectories** | 2006-2016 | | Modelled hourly | | Stein et al., 2015; Draxler and Hess, 1998; (available at: https://www.ready.noaa.gov/HYSPLIT_linux.php) |
| | 2006 | | | Meteorological input: 1°, NCEP/FNL | (available at: https://www.ready.noaa.gov/archives.php) |
| | 2007-2013 | | | Met. input: 1°, NCEP/GDAS | " |

| | 2014-2016 | | | Met. input: 0.5°, NCEP/GDAS | " |
|---|---|---|---|---|---|
| **Condensation Nuclei ($N_{CN}$)** | 2009-2016 | 85% | Hourly median | Instrument: CPC run in parallel with CCNC; Smallest measured particles ~10 nm | Paramonov et al., 2013, 2015 |
| **Cloud Condensation Nuclei ($N_{CCN}$)** | 2009-2016 | | Rounding to nearest hour (or average if multiple) | Instrument: CCN-100, sampling at 8 m | Paramonov et al., 2013, 2015; Roberts and Nenes, 2005; Rose et al., 2008; Schmale et al., 2017. |
| | | 57% (35-73%) | | at $S_{eff}$=0.2 | |
| | | 33% (0-63%) | | at $S_{eff}$=0.5 | |
| **Aerosol number size distribution** | 2006-2016 | 99% | measurement at time of trajectory arrival | Instrument: DMPS sampling at 8 m | Aalto et al. 2001 |
| **In-situ meteorological parameters (T, RH, q, P)** | 2006-2016 | 95-100% | Hourly medians, or sum (P) | Measurements ($T_d$, T, RH, P) at 16-18 m ($p_0$ at ground-level) | available at: https://smear.avaa.csc.fi/; specific humidity (q) derived from $p_0$ and $T_d$ (following equations by WMO, 2018) |
| **Cloud optical thickness (COT)** | 2006-2016 | 5% | Rounding (pixel-weighted average if multiple observations) | MODIS Terra & Aqua Mean or median in 1°x0.5° geographical grid with minimum 20% cloudy pixel coverage | Platnick et al., 2017 |
| **Cloud water path (CWP)** | 2006-2016 | 5% | " | " | " |
| **Cloud Fraction (CF)** | 2006-2016 | 11% | " | " | " |

## 3 Results and Discussion

During the approximate growing season from 1 April to 30 September in the 11 years of observations considered here, 27% of trajectories came from the clean sector. For these trajectories, the mean and median ToL were 38 and 33 hours, respectively, and the mode of ToL rounded to the nearest 5 hours was 25 hours. The average monthly distribution of ToL is shown in Figure 2. In April and September, the median ToL between all the years considered were only 28 and 31 hours, while in August it was exceptionally long (44 hours) in comparison to the rest of the months. Clean sector trajectories were, however, also noticeably rarer in August, and the relative spread of the observations is larger.

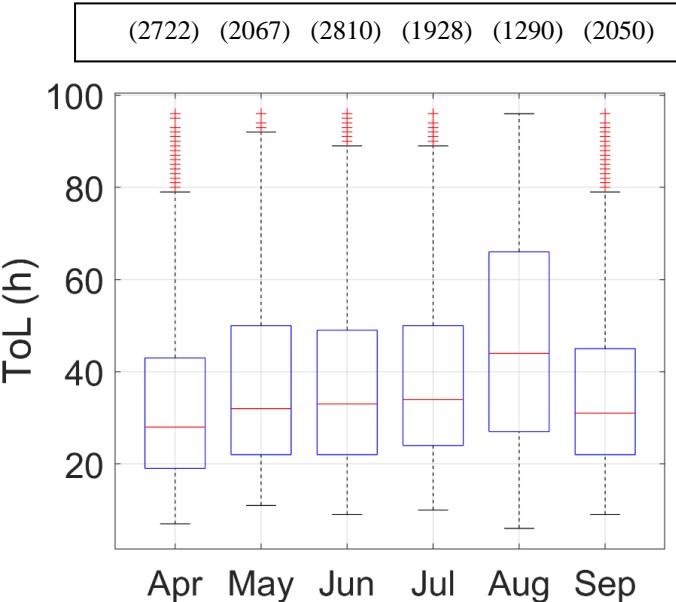

Figure 2. Monthly (growing season) distribution of ToL in air masses arriving from the clean sector in 2006-2016. The red bars show the median of the observations, while the top and the bottom of the boxes indicate the 75th and 25th percentiles. The whiskers mark the furthest observations that are not classified as outliers, which are indicated with the red crosses and are more than 1.5 times the interquartile range away from the box limits. The number of clean sector trajectories in each month in the 11 years is shown in the text box above.

### 3.1 Aerosol particles as a function of Time over Land

In Figure 3, arriving air masses are divided according to their aerosol particle number concentration (measured with a CPC and therefore termed here the number concentration of condensation nuclei, $N_{CN}$) into 4 groups that each represent a quarter of all observations, making the group limits quartiles. The fraction of air masses belonging to each value group changes with ToL. In air masses with the shortest ToLs of approximately 10 hours, condensation nuclei concentrations are mostly low: nearly half of them have number concentrations below the lower quartile (1400 cm$^{-3}$), and more than 60% concentrations less than the 50$^{th}$ percentile, i.e. the median (2200 cm$^{-3}$). Air masses with slightly longer ToLs tend to have much higher $N_{CN}$. Less than 15% of air masses that had spent 20 hours over land, belonged to the group with concentrations under the lower quartile (1400 cm$^{-3}$). Nearly 45% of them on the other hand, belonged to the 75$^{th}$-100$^{th}$ percentile high concentration group, with number concentrations exceeding 3400 cm$^{-3}$. The high abundance of particles at these relatively short continental transport times is likely a result of the very high occurrence rate of NPF events in such air masses (Petäjä et al., 2022).

It appears that In a relatively short exposure (15-20 h) to the forest, an originally marine air mass accumulates enough BVOCs and subsequent oxidation products to drive NPF (Tunved et al., 2006; Petäjä et al., 2022). Why the characteristic total number concentration is lower at the shortest 10 hours of ToL could be explained by lesser BVOC accumulation not being able to promote as much NPF, but also by NPF inhibition resulting from cloudiness, as these fast-moving air masses may be more likely to be associated with cloudy frontal activity. When ToL increases beyond 20 hours, the point where highest number concentrations were observed, we can increasingly expect the air masses to have already undergone NPF and have a population of pre-existing particles that available condensable vapours are more likely to partition onto, so that the probability for further NPF is decreased. At the same time, existing particles are subjected to sinks like coagulation or deposition. Correspondingly, we observe that the probability for high number concentrations decreases between 20 and 60 hours of ToL, concentrations above the median 2200 cm$^{-3}$ for example dropping from the peak value of 70% to only around a third of cases. The decrease in the fraction of observations belonging to the larger value groups is the most rapid roughly between 20 and 50 hours, after which it slows down. A Pettitt's test fitted to bin medians starting from the 20-hour bin (ignoring the opposite trend at shortest ToLs), agrees with visual interpretation and also suggests a change point in the 50-hour ToL bin, although the trend still remains slightly negative (for example, a least-square linear regression fit to observations still has a slight negative slope after this point). However, the evident slowing down of the differences between the distributions of $N_{CN}$ between the ToL bins after approximately 50 hours suggests that after this time period the aerosol number concentration sources and sinks gradually start to advance a balanced state within the forest environment, although a fully static state does not seem to be reached in this time scale and could still be further down the line.

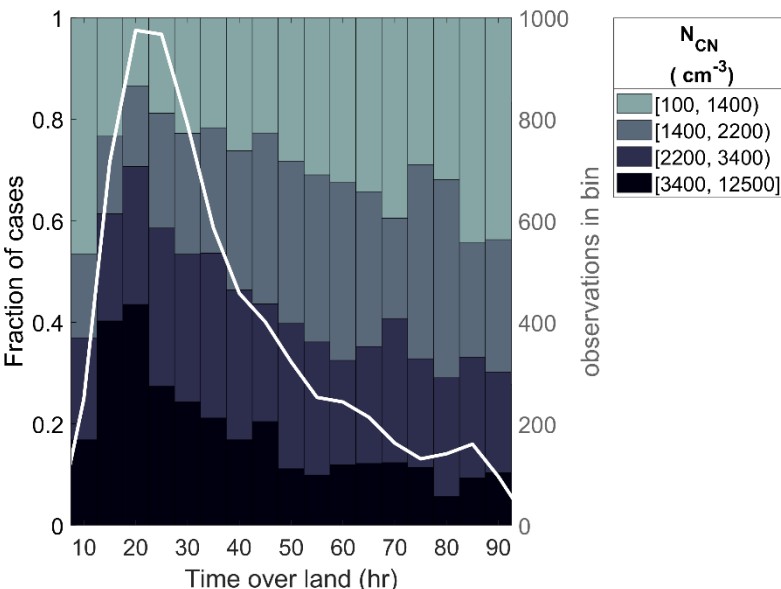

**Figure 3. Fractions of measured condensation nuclei ($N_{CN}$; sized ~10nm and larger) number concentrations in 4 different value groups in air masses divided into 5-hour Time over Land bins. The white line and the right y-axis show the number of observations in each bin. Six outliers more than 6 mean absolute deviations (MAD) from the mean were excluded (between 12500 and 19000 cm$^{-3}$). The ranges are based on data percentiles, the value group roughly representing the 0th-25th, 25th-50th, 50th-75th and the 75th-100th percentiles. Note, that this means that the range is not equal between groups.**

Figure 4 provides additional information on also the size of the particles, by showing the median aerosol number size distributions in the same ToL bins. The distribution closely resembles the evolution of aerosol number size distribution during a regional NPF event (see e.g., Dal Maso et al., 2005). A Similar dependence of aerosol number size distribution on ToL, has also been observed in earlier studies (Tunved et al., 2006; Petäjä et al., 2022). The relative frequency of similar particle formation and growth processes that take place in a single location during an event, seem to dominate and determine the average aerosol population in air masses with increasing times of exposure to the boreal forest. In the first few bins in Figure 4, the number distributions peak at relatively small sizes, as would be expected from air masses where many of the particles have formed fairly recently in NPF events. The highest concentration peak overall is seen in the 20 h ToL bin, in agreement with Figure 3. Increase in ToL is accompanied with a shift of the number distributions to larger particle sizes, in line with our hypothesis that these particles have been growing in the forest environment. Eventually, the number distribution peak appears to settle to a seemingly consistent size when ToL reaches approximately 55 hours (after which, any further deviation is only up to a few percent). Overall, between the 10-hour and 55-hour ToL bins, the peak of the median distributions shifts from approximately 22 nm to 72 nm, corresponding to an average rate of change of roughly 1.1 nm per h of ToL.

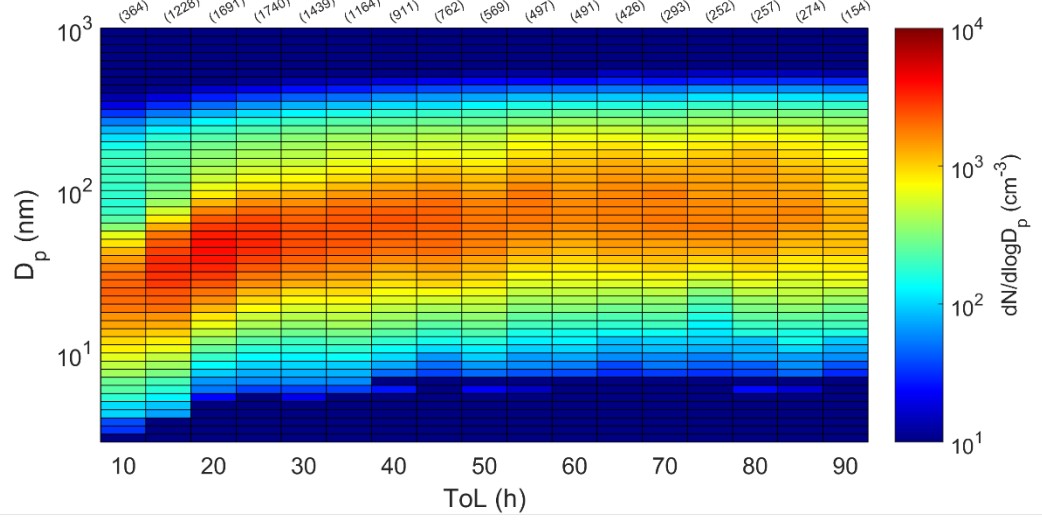

**Figure 4. Median aerosol number size distribution in air masses with different ToL. The numbers in top show the number of trajectories in each bin.**

While Figure 4 shows the growth and shift of particle size distributions towards larger sizes between 10 and 60 h ToLs, from Figure 5 we can see a corresponding increase in measured CCN concentrations. We analysed CCN measurements at two supersaturations, 0.2% and 0.5%. In Hyytiälä, the median (non-normalised method) critical diameters at these supersaturations are 96 and 67 nm, respectively (Paramonov et al., 2013; the latter value estimated from the values provided therein). In clouds, peak supersaturations vary greatly depending on the aerosol load and meteorological conditions like the updraft velocity. For example, if the air is polluted, effective supersaturation in a stratus cloud may be approximately 0.1%, while in clean maritime air mass it can regularly exceed even 1% (Hudson and Noble, 2014). In Hyytiälä, 0.2% supersaturation in clouds can be expected to be quite common, while 0.5% might require high vertical velocities and a relatively low numbers of CCN (Pruppacher and Klett, 2010; Pinsky et al., 2014; Väisänen et al., 2016).

The differences in CCN concentrations between the air masses with different land travel times are notable. At the supersaturation of 0.2% for example (Figure 5a), when ToL is only around 10 hours, 70% of air masses have $N_{CCN}$ less than 110 cm$^{-3}$, belonging to the 0[th]-25[th] percentile group. In both the 10 and 15-hour bins, only approximately 10% of observations have numbers higher than the median value of 180 cm$^{-3}$. After this, there is an increased probability for higher CCN number concentrations until about 55 hours of ToL, after which the probability for specific CCN ranges appear to be less influenced by ToL. The loss of relationship after this point was confirmed by a Pettitt's test on the 20 to 90-hour ToL bin medians, which found a change point in the 50-hour ToL bin. A least-squares linear regression line fitted to observations approximately levels off when fitted from the 55-hour bin onwards (to observations with ToL 52.5-92.5 h), being also a clear indication of no further relationship. Therefore, we can interpret that the air masses reach a relatively balanced state with regards to CCN sources and sinks in the new environment by around 50-55 hours ToL. In the ToL-bins after this point, around 95% of cases exceed the first quartile (110cm$^{-3}$), while around 80% have number concentrations higher than the median (180 cm$^{-3}$), and more than half

have number concentrations exceeding the third quartile, being between 300 cm$^{-3}$ and 1660 cm$^{-3}$. Remembering that 0.2% supersaturation can be considered relatively typical, these CCN can be expected to have a reasonable potential to activate as true cloud condensation nuclei in the environment, potentially affecting cloud properties.

When measuring CCN at 0.5% supersaturation, a wider range of particles is activated and we see higher particle numbers, but otherwise a similar development with increasing concentrations at longer ToLs (5b). Here, the fractions of higher concentrations again cease to consistently increase after a point. A Pettitt's test and least-squares regression line fitting executed similarly as before, suggesting a change-point in the 45-hour bin, and a levelling off of least-square fit starting from the 50-hour ToL bin observations. Examining Figures 5 c and d demonstrates that the increase in CCN concentrations results largely from the increased $N_{CCN}$ to $N_{CN}$ ratio. Particles grow in size while travelling in the forest environment, and consequently more and more of them enter the CCN size ranges. Here, it is again visually observable that the fractions change much less somewhere after 50 hours ToL, around 55-60 and 50-55 hours for 0.2% and 0.5% supersaturations, respectively, suggested by Pettitt's testing and least-squares regression fitting. These findings agree with earlier observations of increasing CCN number concentrations with ToL, made in the same location (Petäjä et al. 2022), while an increase in a related property of total aerosol mass, has also been observed both in the same location (Tunved et al., 2006; Petäjä et al., 2022), and also in a coastal arctic Russian site (Asmi et al., 2016).

(a)

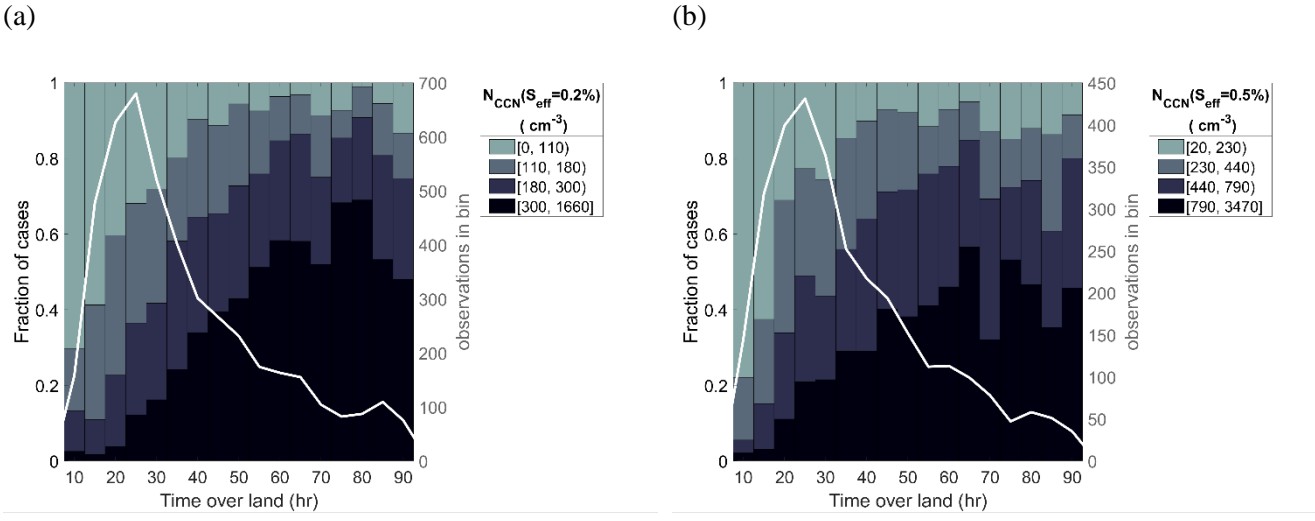

(b)

(c)

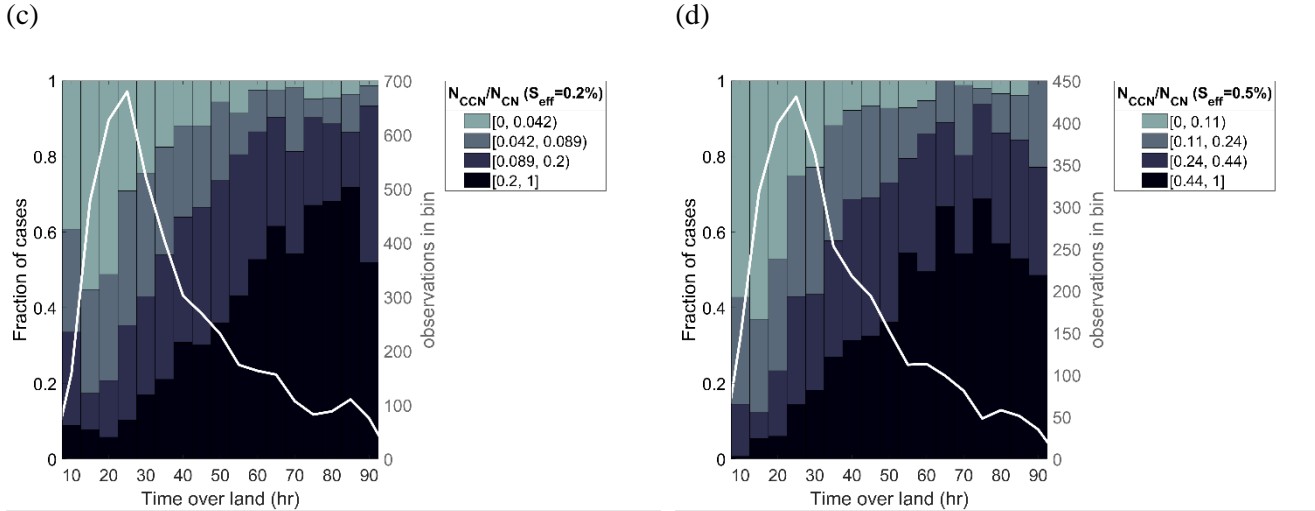

(d)

**Figure 5. Fractions of observed (a) N$_{CCN}$ and (b) N$_{CCN/CN}$ ratio at 0.2% supersaturation, and (c and d) at 0.5% supersaturation in 4 different value ranges, as a function of ToL. The white line and the right y-axis show the number of observations in each bin. The general description of this figure is the same as for Figure 3. Fourteen and five outliers (MAD>6) were excluded from figures (a) (1700-3900 cm$^{-3}$) and (c) (4300-7200 cm$^{-3}$) respectively.**

## 3.2 Atmospheric humidity as a function of Time over Land

Air masses with longer ToL tend to contain more water vapour (Figure 6a). While high specific humidities (q) above the upper quartile of all observations, between 6.3 and 13 g/kg, are nearly non-existent at the shortest ToLs, they constitute around 40%

of the air masses with longer ToLs (of around 60 hours or more). Conversely, very dry air masses with specific humidities below the first quartile (the 0-3.5 g/kg group) account for approximately 40% of the observations when ToL is very short (15 hours) but are much less common (~15%) when ToL is long. Fraction of observations above median (5 g/kg) increases from about 25 to 65% between short and long ToLs. The figure again shows that the grouping of air masses into the 4 different ranges, changes consistently at shorter ToLs, but the trend begins to diminish when the land transport times get longer. To support what can already be visually seen, a Pettitt's test on bin median q values (for 20-90 h bins) locates a change in central tendency starting from 50 h ToL, and with a slow balancing, a completely flat least-squares regression line fits observations with at least 62 h ToL, suggesting that at least by this point there should not be a relationship between ToL and q.

The differences in specific humidities between air masses are at least in part explained by the temperature differences (Figure 6b). In Figure 7, the plot of simultaneous observations of specific humidity and temperature provides a reminder of the clear maximum boundary the saturation curve imposes to the specific humidities at any given temperature. This illustrates how, for example, specific humidities that are above 6.3 g/kg (i.e. the upper quartile) are only possible when temperatures are above 6.6°C. More than 65% of air masses with very short ToLs of 10 hours and less are colder than this (not shown). Typical air mass temperatures increase rapidly in the relatively short ToL range, and when ToL reaches 15 hours, already more than 30% of incoming air masses are warmer than the median of all observations, 9.8°C. When ToL is above roughly 51 hours (where both a Pettitt's test gives a change-point and from which onwards a flat regression line fits observations), fractions of air masses within the different temperature ranges already seem relatively constant, and approximately 80%, 60% and 35% exceed first (5.5°C), second (9.8°C) and third (14°C) quartile temperatures, respectively.

This is in line with our expectations that in summer, the air masses warm up as they travel over land, as more energy partitions into sensible heat flux over the continent than over the ocean (Wild et al., 2015). Our studied season, however, covers half a year, so the temperature variation is significant also from seasonal effects alone, which might interfere with separating the effect of the land transport from possible seasonal differences in typical atmospheric flow. For example, the average of the ToL during the colder months (April and September) was shorter than in rest of the months (especially August, which however had much less clean trajectories and a highest relative variability) (Figure 2). Therefore, we also compared the individual observations to the monthly median temperatures of the 11-years (of all air masses from all directions), to confirm the warming associated with land transport (Figure 6c). The median temperature difference was approximately -1°C, indicating that the air masses from the north-western sector considered here, are slightly cooler than the average between all air mass directions. The warming of air masses with longer travel times over land can also be seen here. After the local minimum at 17-hour ToL, where even 80% of air masses were colder than the median deviation, the fraction of warmer air masses increases with ToL. Finally, after approximately 50 hours there were consistently mostly a little more air masses warmer than the median, than air masses colder than the median. (More specifically, a Pettitt's test places a change point to 52 h, and a flat regression line can be fitted to observations from 53 h ToL onwards.)

The warming of the airmasses as they travel over land has a key role in enabling an effective uptake of the water vapour, thus facilitating the increasing specific humidities (Figure 6a), as it strongly increases the vapour-pressure deficit and the possible

amount of moisture the air can hold (Figure 7). The water itself is likely to originate from forest evapotranspiration (Hornberger et al., 2014), which underline the importance of forests as a water source. In line with our hypothesis of increasing specific humidity being mostly facilitated by warming, no clear trend is seen between relative humidity (RH) and ToL, only some
fluctuation (Figure 6d). This might suggest that the evapotranspiration from the forest provides a vapour source strong enough to sustain relatively similar RH distributions in the air masses with longer ToLs and (on average) higher temperatures but is not so strong that it would also shift the RH distribution.

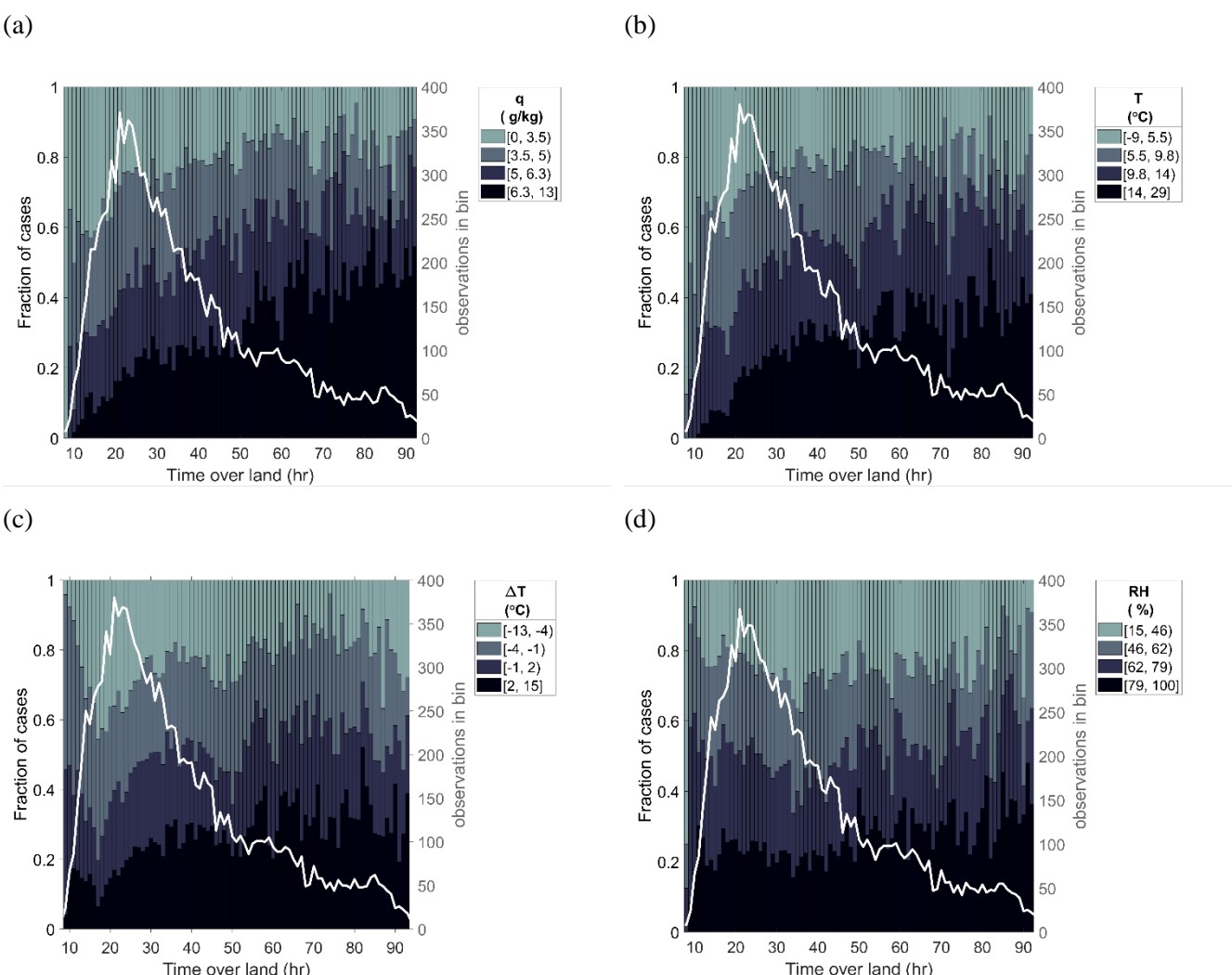

**Figure 6. Fractions of observed (a) specific humidity (q), (b) temperature, (c) temperature deviation from monthly median and (d)**
**relative humidity, in 4 different value ranges, as a function of ToL. The white lines and the right y-axes show the number of observations in each bin. The general description of this figure is the same as for Figure 3, except that here the ToLs are binned into hourly bins, which was possible with the higher data availability. The same ToL cut-off limit of 92.5 h is still maintained for consistency.**

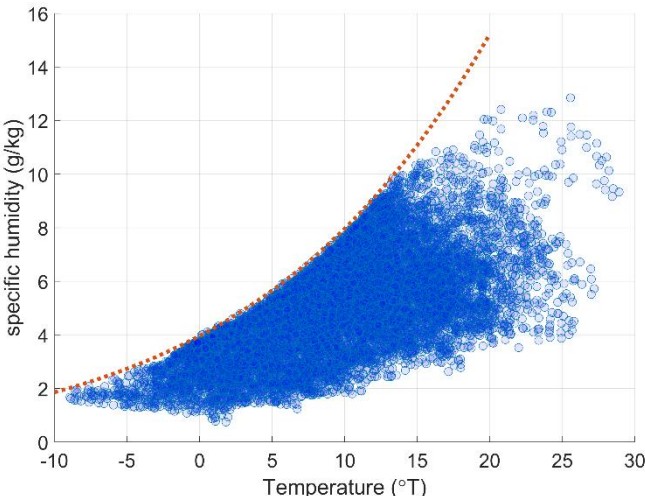

**Figure 7. Observed specific humidities plotted against simultaneous observations of temperature. The red line shows the saturation specific humidity at the lowest hourly average atmospheric pressure of the observations (962 hPa).**

### 3.3 Cloud observations as a function of Time over Land

Next, we compared satellite-based cloud observations over Hyytiälä to the air mass ToL, in order to examine potential responses in connection with the processes already observed at the surface. Previous shorter-term investigations have already suggested increases in cloud droplet number concentrations of liquid clouds as a function of ToL (Petäjä et al., 2022). Here, we investigated satellite-retrieved observations of cloud cover and optical properties in a $1° \times 0.5°$ (latitudinal $\times$ longitudinal) sized grid enclosing the SMEAR II station. Only satellite images reaching the criterion of minimum 20% relevant pixel coverage were considered. The data set is limited, as coinciding satellite images and clean-sector trajectories are required. Median Cloud Optical Thickness (COT) within the grid is shown in Figure 8a. As the COT is estimated from 1 km $\times$ 1 km pixels that are expected to be overcast, the focus is mostly on relatively uniform cloud covers (e.g. stratus type clouds and stratocumulus). In comparison to many of the earlier figures, no very clear or persistent trend is seen, and interpreting and drawing conclusions from the figure are somewhat challenging due to the limited number of observations (see the white line and corresponding y-axis in figure). A modest increase in observations with higher median COT values seems to however occur after ToL exceeds approximately 50 hours. A t-test fitted to log-transformed COT data also suggests that the distribution of observations going into the bins with ToL 45 hours or less are at least statistically significantly different ($p<0.05$) from the distribution of observations in the 50-90 h bins. In the bins with ToL 50 hours or longer, mainly the fraction of observations exceeding the upper quartile, COT>16, seem to be momentarily about half more common than at shorter ToL bins. At around and after 80 hours of ToL, lower median COT observations however again suddenly take up a larger proportion of the observations for an unknown reason. However, the number of datapoints in the bins with longer ToLs are low (see the white line and the right y-axis in the figure), so the groupings are likely to be less reliable.

Mean COT values of the satellite image pixels within the 1° × 0.5° area (Figure 8 b) differ from the median values (Figure 8 a). The mean cloud optical thickness can give more emphasis on thicker smaller clouds that deviate more from the median of the whole grid. As a result, a patchier cloud layer with occasional thicker regions may have a much higher mean than median COT, explaining the higher mean values in Fig. 8c. Although the mean COTs are higher in value than median COTs, their
behaviour as a function of ToL does not seem to differ from each other, as we see similar increases after roughly 50 hours. The lack of major differences between the mean and median COT might indicate that there is little change in the patchiness of the cloud cover with ToL; at least not enough to be observed at this resolution, and especially with the limited number of datapoints.

A hint of a somewhat similar phenomenon can possibly be seen in the cloud fraction (CF) (Figure 8c), where the clearest skies
(median CF<0.14 and mean CF<0.24) seem to become a little less common when ToL is longer than 50 h. More than anything, there seems to be a slight drop in the prevalence of cloud fractions higher than these (i.e. median CF>0.14 and mean CF>0.24) somewhere between 35 and 50 hours but inferring any trends beyond this may not be justified, although overcast views seem also a little more common between 55 and 75 hours of ToL. The shortest ToL bin is again an exception, where cloud fraction is much higher than elsewhere, most likely an effect associated with frontal activity. Unfortunately, we were unable to confirm
statistical significance of the differences in CF with the limited data.

The figures discussed above suggest that the effects driven by land transport onto the near-ground properties may also translate into the cloud level at relatively similar time scales. These results should, however, be considered only indicative, as the number data points is limited, making the effects quite uncertain. One should also remember that our data only include overcast pixels, which can favour certain cloud types, and our cloudy pixels may include different cloud phases, while additional
uncertainties may also arise from e.g., thin clouds and sub-pixel inhomogeneities (Zhang et al., 2012). Therefore, at this point, we merely consider these as an example of possible observable effects in the cloud level and contemplate that a more detailed future analysis, possibly involving ground-based measurements with a much higher resolution, would be beneficial in confirming any possible effects.

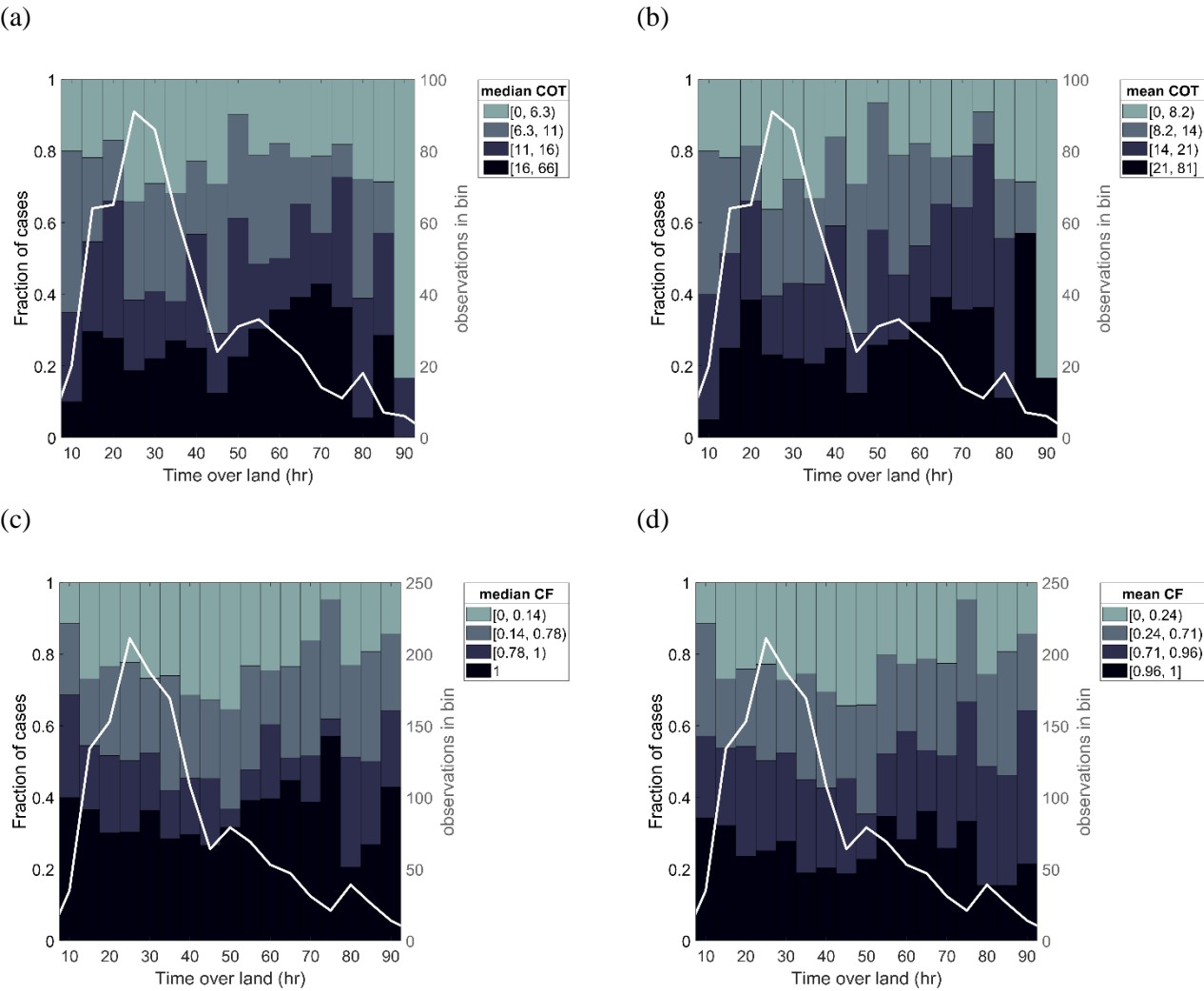

**Figure 8. Fractions of satellite-observed 1°×0.5° grid (a) median COT, (b) mean COT, (c) median CF and (d) mean CF in 4 different value ranges as a function of ToL. The white lines and the right y-axes show the number of observations in each bin. The general description of this figure is the same as for Figure 3. COTs are only considered from satellite views where the number of expected overcast pixels in the 1°x0.5° area is at least 20% of the maximum observed. For CF all observations where the partial view over the area is 20% or more are accepted. Note that in (a) and (b), the two last ToL-bins have only 9 and 6 observations respectively and are therefore very unreliable.**

## 3.4 Precipitation as a function of Time over Land

Precipitation also varied with ToL. In Figure 9, the precipitation accumulation is similarly divided into value bins and their fractions are shown against the corresponding ToL. Figure 9a depicts the accumulation of longer-term precipitation from the next three hours after the arrival of the air mass. With the exception of the very short ToLs (≤12 h), with a high probability for

frontal precipitation, the rain events increase in probability with an increasing ToL. This increase is modest and not strictly monotonic, while somewhat similarly to the cloud observations, the precipitation probability seems to be higher especially after a land transport time of roughly 50 hours. Any possible stabilisation of fractions is difficult to infer here. In slowly moving air masses, it is also possible, that higher precipitation accumulations could be partially due to same clouds precipitating longer times over the same location. A drawback of investigating the 3-hour precipitation with hourly trajectories is that the rainfall (or lack thereof) in one specific hour could be sampled up to 3 times (for 3 subsequent trajectories) if the air mass source region remained the same for an extended period. However, we also see a similar increase in precipitation after approximately 50 hours of ToL, in a shorter, 1-hour precipitation accumulation, that is free from this issue (Figure 9b). We conducted Pettitt's test on the mean values (not medians as precipitation data is dominated by zeroes) of bins starting from 13 h of ToL for both precipitation accumulation periods, to see if it would also suggest a change-point matching the visual estimation. The test located a statistically significant change-point for the 1 h precipitation accumulation to the 48-hour bin. The precipitation probability (P>0) in the 1 hour after an arrival of an air mass was on average 7% in the 13 to 47-hour ToL bins, and 12% in the bins with longer ToLs. For the three-hour accumulation case, the increase after around 50 hours appears less robust as no statistically significant change point was identified (only a marginally significant (p~0.07) changepoint at 55-hour ToL bin, with 12% and 18% average precipitation probabilities before and after). Flattening least-square fits were not found, but for all observations in the ToL range of 12.5-92.5 h ToL, a statistically significant least-square linear regression line with a modest positive slope fit observations of both rain accumulation periods, also indicating an overall positive trend between precipitation and ToL.

(a)                                                    (b)

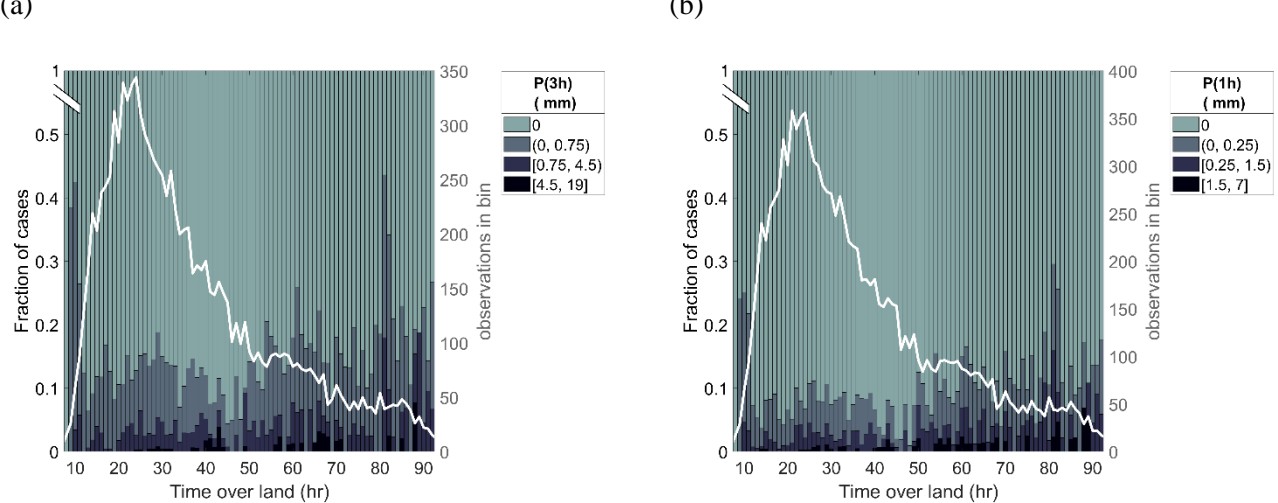

**Figure 9. Fraction of rainfall in the next 3-hours (a) and 1 hour (b) in 4 different value ranges as a function of ToL of the arriving air mass. There is a break in the y-axis so that distinguishing the detail of precipitating fractions is easier. The white line shows the number of observations in each bin. Here the value groups were manually selected and not based on percentiles. The rainfall value ranges are also different between the figures, the 3-hour precipitation limits being triple the 1-hour limits. One outlier was excluded from (b) (18 mm).**

The increased precipitation probability with an increasing ToL is in line with the previous observations of both specific

humidity and clouds. The evapotranspired water vapour appears to be efficiently spread into the atmospheric column, where

it can condense onto cloud droplets. This is demonstrated by Figure 10, which shows a positive correlation between the surface-

measured specific humidity and the cloud water path (CWP) measured at the same time in the atmospheric column. Similarly

to other satellite-derived variables, the shown CWP is a median of a $1° \times 0.5°$ grid, when the grid had at least 20% cloudy

pixel coverage. A high CWP is seen especially in connection with specific humidities above 6 g/kg, although the scatter is also

quite large at these higher values. These high specific humidities are more common in air masses with longer ToLs (Figure 6),

so the associated increase in CWP can also explain the observations made between COT and ToL (Figure 8 a&b).

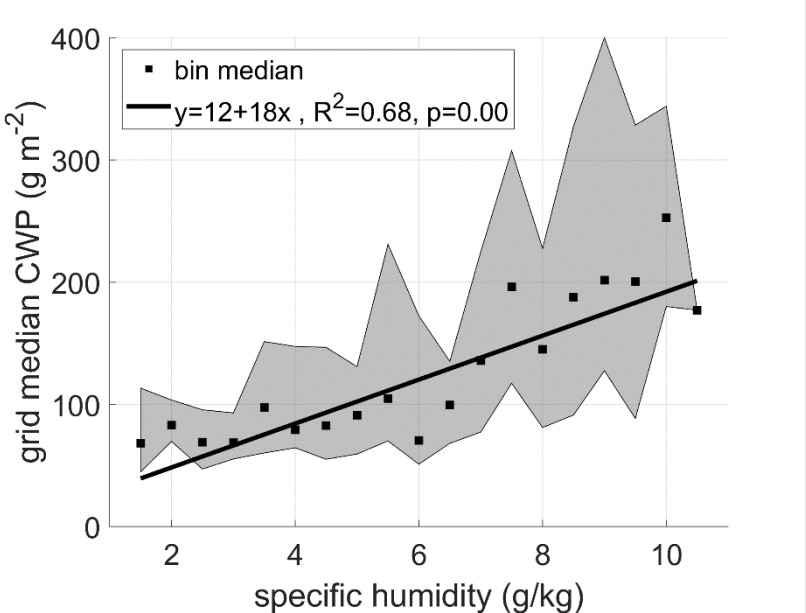

**Figure 10. Cloud water path as a function of specific humidity. The squares correspond to the median of observations in each 0.5 g/kg-wide specific humidity bin, and the range between 25th and 75th percentiles is shaded. Individual satellite observations were median of CWP observations in a 1x0.5-degree view over Hyytiälä, when the cloudy pixel number was at least 20% of the maximum (i.e., same criterion as for the considered COT observations in Figure 8). A least-squares regression weighed with the number of observations in each bin is fitted to the bin medians . The regression equation, adjusted $R^2$ and p-value are shown in the legend.**

Ultimately, the moisture build-up with longer ToL is expected to explain some of the higher precipitation frequencies (Figure

9). A linear trend between specific humidity and precipitation rate has been previously observed throughout northern Eurasia

(Ye et al., 2015). In Figure 11, we have plotted the mean precipitation in the hour following the arrival of an air mass from our

selected source region against the specific humidity measured at the station in the beginning of the same hour. A clear positive

trend is present when specific humidity is in the range of 3 to 9 g/kg. Precipitation below a "threshold value" of 3 g/kg appears

to be minimal. Whereas, the specific humidity of 9 g/kg is linked with an especially high mean precipitation of approximately

0.25 mm, averaged over roughly 160 measurements (see grey line). Specific humidities higher than this were relatively few in

numbers (less than 2% of observations) and did not seem to be associated with similarly high precipitation. Perhaps precipitation is too improbable, especially heavier rain, and the low mean values in these bins is simply down to random variation, since even the 9.5g/kg bin has only 70 measurements (17 of which have any precipitation). When focusing on rain
events specifically ("wet hours", Figure 11b) for a closer look on the intensity of the precipitation, there also was a slight observable trend between median rainfall and specific humidity, despite the variance being large especially at higher humidities. Therefore, in addition to observing a link between specific humidity and precipitation frequency, we see that higher humidities can also be associated with higher precipitation rates, when precipitation does occur.

(a)                                                                                              (b)

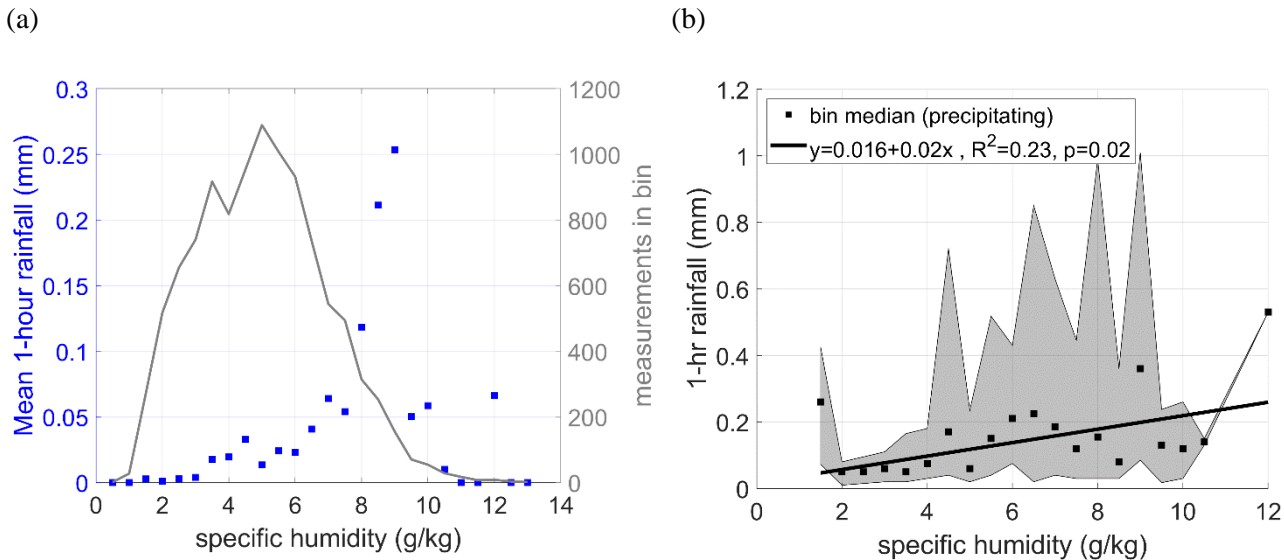

**Figure 11. (a) Mean of rainfall (blue squares) accumulated in an hour (including both precipitating and non-precipitating cases), as a function of specific humidity measured at the beginning of the hour. The number of data points in shown again by a grey line. (b) Median precipitation accumulated in single "wet hours" as a function of specific humidity measured at the beginning of the hour. A least-squares regression weighed with the number of observations in each bin is fitted to the bin median data and the regression equation, adjusted $R^2$ and p-value for the fit are shown in the legend. One exceptionally high precipitation outlier was omitted from**
**both figures.**

## 4 Conclusions

We investigated the influences of boreal forests on clean air masses entering the Fennoscandian land area from the ocean between northern and western directions, and explored how these initially marine air masses transformed as they travelled over land. Specifically, we focused on analysing how in situ aerosol particle number size distribution, cloud condensation nuclei
concentrations, humidity, cloudiness and precipitation were influenced by the interaction time between the air masses and the boreal biome. We followed the approach of e.g., Tunved et al. (2006) and Petäjä et al. (2022), and determined a Time over Land (ToL) value for hourly air mass back trajectories located within our assigned clean sector. Our analysis covered an extensive eight to 11 years of data, between the months of April and September, from the SMEAR II station in Hyytiälä,

Finland. The utilised long data sets bring added confidence to the results and support earlier investigations (e.g. Petäjä et al.,
2022) focused on considerably shorter study periods.

Our analysis showed major differences in air mass properties with different times over land. In line with previous observations of NPF frequencies (Petäjä et al., 2022), high number concentrations were often observed especially around 20 hours of ToL. When air masses had experienced a longer ToL, their total particle number concentrations were typically lower. Having had had time to grow larger in the presence of condensable vapours produced in the forest environment, the remaining particles
were however larger, and significantly higher concentrations of CCN were seen in air masses with longer ToLs. In their transition from marine (short ToL) to forest-continental (long ToL), air masses had also typically accumulated more water vapour, made possible by the associated warming of the air masses, and the moisture provided by plant evapotranspiration. Higher specific humidities at the surface were connected to higher column cloud water paths and increased precipitation, suggesting that the moisture provided by the forest environment can propagate to the cloud level, explaining some of the
observed changes in cloudiness and increased precipitation frequency that were also seen with longer ToLs. While evapotranspiration can provide the water content necessary for cloud formation, the accumulation of CCN with longer ToLs could potentially lead to an increased number concentration of cloud droplets (observed previously by Petäjä et al., 2022), which could boost the reflectance of the cloud cover (Christensen et al., 2020; Yli-Juuti et al., 2021). Correspondingly, we did observe a small potential increase in the frequency of higher COT cases after longer ToL, but due to a small effect on a
relatively small amount of data, confirming these findings calls for further research. These observations are however in line with the previous findings by Petäjä et al. (2022), where they found also higher cloud droplet number concentrations and liquid water paths in air masses with longer over-land transport time, which they hypothesised could translate into higher cloud optical thickness.

Approximately 50-55 hours of land transport arose as a significant time scale in our observations.  After this time, for many
of the variables, the distribution i.e., the fraction of observations in the four value groups, remained relatively stable and did not change significantly with further increase in ToL. Therefore, it appears that in this is an approximate timescale in which an originally marine air mass transforms into an air mass that is in a relatively balanced state with the boreal forest environment in terms of sources and sinks of aerosol particles and moisture. Specifically, CCN (measured especially at 0.2%) and the provided moisture are key variables for cloud formation, and consequently, changes were also observed around a similar
timescale in the cloud level. Slightly elevated cloud optical thicknesses and cloud fractions, and an increase in precipitation frequency, were observed after air masses had travelled over land for 50-55 hours. However, as mentioned previously, findings relating to clouds were not statistically very robust and need more research. In their relatively similar analysis based on a much smaller data set, Petäjä et al. (2022) concluded that air masses experience changes in aerosol and cloud properties for up to 3 days of transport over land. In on our analysis, this time scale appears to be shorter, as we suggest that around 55 hours seems
to be sufficient for the aerosol-cloud interactions driven by the boreal forest emissions to reach their full effect. This can also be adapted to reflect a required size of the area, if considering average wind speeds. This transitional time frame observed, highlights the fact that observations made closer to the boundaries of the boreal forest may not be generalisable over a wider

region, as in many of the air masses in such locations, the forest interactions might not have yet taken their full effect. At SMEAR II for example, the median ToL of clean sector trajectories was 33 hours, i.e., shorter than the suggested time of saturation of forest-aerosol-cloud effects, and therefore e.g., typical average CCN concentrations measured there, might not be representative of the typical concentrations at a location where the median would be 60 hours. Our investigation covered observational evidence from the first four days of forest-air mass interaction. How the interactions may develop later down the line, is however something we cannot speculate with any clear certainty.

Our results demonstrate boreal forest potential to form clouds by promoting higher absolute humidity and CCN formation. The observed moisture provided by the forest, and the increase in precipitation with ToL as air masses transform from marine to continental over the boreal forest, highlights the humidifying role of forests. We expect that a similar tendency towards more cloud seeds, humidity and precipitation is likely also elsewhere, over other forest-covered northern edges in Northern Eurasia and North America, when air masses transform from polar marine to continental air masses. However, the link to cloud formation and changes in cloud properties still remains somewhat obscure due to the relatively low number of investigated satellite observations, even within the rather long time-interval of 11 years and will therefore remain an open question for possible future analysis.

**Code and Data availability:** Several data sets are available through the references provided in the manuscript, and the remainder upon request from the corresponding author.

**Competing interests:** The authors declare that they have no conflict of interest.

**Author contribution:** The idea and design of the study were conceived by EE, TP and MK. MR wrote the manuscript, analysed most of the data and provided the visualisations under the supervision of EE and TP, while also LS, VMK, TN and MK were involved in the interpretation. Supporting analysis or data were provided by HK and LS. All authors also contributed to the manuscript through reviewing and commenting.

**Acknowledgements:**

The work was supported by Atmosphere and Climate Competence Center (ACCC) Flagship funded by the Academy of Finland (grant number 337549), Center of Excellence in Atmospheric Sciences, and Academy of Finland with several projects, such as NANOBIOMASS (grant No. 307537), ACRoBEAR (grant No. 334792), and grant number 311932. The work was partially funded by European Commission via iCUPE (grant No. 689443) and FORCeS (grant No. 821205) projects. We also acknowledge the projects: "Quantifying carbon sink, CarbonSink+ and their interaction with air quality" INAR project funded by Jane and Aatos Erkko Foundation, and European Research Council (ERC) project ATM-GTP (contract No. 742206).

We gratefully acknowledge the NOAA Air Resources Laboratory (ARL) for the provision of the HYSPLIT transport and dispersion model used in this publication. We are grateful for Pasi Aalto for providing the automated trajectory calculation utilized in this work. We extend our gratitude to the SMEAR II technical staff for their maintenance work ensuring good quality data during the last decades. Raw data were generated at INAR, University of Helsinki. The MODIS/Terra and Aqua Clouds 5-Min L2 Swath 1km and 5km datasets were acquired from the Level-1 and Atmosphere Archive & Distribution System (LAADS) Distributed Active Archive Center (DAAC), located in the Goddard Space Flight Center in Greenbelt, Maryland (https://ladsweb.modaps.eosdis.nasa.gov/). We also thank the European Forest Institute (EFI) for providing the data for the Forest Map of Europe.

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
