# Peer review of "Dynamics of aerosol, humidity, and clouds in air masses travelling over Fennoscandian boreal forests"

_Atmospheric Chemistry and Physics, 2022_

## Author Comment (AC1)

We would like to thank both referees for their feedback. The comments are copied in this document and our responses are written in blue under each comment.

References are provided alongside the comments for citations that were not in the reference list of the preprint.

**Anonymous Referee #1**

The authors use observed cloud properties and precipitation events in conjunction with airmass back trajectories to explore the effect that a Finnish boreal forest has on aerosol-cloud interactions, which are a key, potentially cooling, process in the climate system.
This is a nicely conceived study that builds well on Petäjä et al (2022), presenting a wider range climate/weather variables and characteristics and supporting previous work on the importance of northern high latitude forests to the climate system. The analysis uses a long time series (11 years) of meteorological and satellite data, and focuses on air masses from the N and W which pass over only boreal forest between the Scandanavian coast and Hyytiälä. The conclusions are relatively robust and relevant, although I have one or two comments about their choice of how to bin the data. Overall this study re-emphasises the importance of boreal forests for future cloud nucleation and hence climate cooling.
I recommend the manuscript be accepted subject to minor revisions to address my concerns, outlined below.

Thank you to referee#1 for their comments, our responses can be found below.

**R1.1**      Abstract: Would benefit from the inclusion of some quantities to make the scale of the issue more apparent.

We agree, now some quantities relating to our results have been added.

**R1.2**      Introduction: The same is true of the first 3 paragraphs of the Introduction which are highly generalised "accepted truths" rather than hard facts and figures.

This is a good point, however, as the second reviewer pointed out (R2.1), many of these topics are not very relevant from the perspective of our study (beyond being a motivation), and this setup discussing climate change and feedback effects, may cause the reader to expect insight specifically related to climate change. Therefore, we have deleted most of the text referred here, and edited the introduction with an aim to better reflect the forest-atmosphere interaction that are investigated in this paper.

**R1.3**      L40-45: please quantify the proportion of the Earth system's carbon, water and energy cycles that flow through boreal forests.

As mentioned, most of the discussion on these topics have been deleted. We think it is still fair to leave the referenced sentence "Conversely, the boreal forest also influences the climate on both local and global scales through the exchange of energy, carbon, water and through the production of atmospheric aerosol" as it is, just to emphasise that the forest has many ways of interacting with the atmosphere, before moving on to discuss the more of processes we are focusing on. (Especially, since there are then a few references provided.)

Providing all-encompassing estimates for these very complex exchanges would also be very difficult task, as they consist of so many smaller elements.

The forest water exchange is however a relevant part of this study, and the role of the boreal forest in the hydrological cycle has recently got an increased attention (Goeking and Tarboton, 2020, Wei et al., 2022).

But again, quantifying the whole cycle is difficult, but here are some examples on the link of boreal forest change and the hydrological cycle:

A recent modelling study shows that the annual runoff response to 5% change in boreal forest parameters (see reference for details) is at 0.75%, which is larger than the average 0.66% for all forest biomes (Wei et al., 2018). Most often, deforestation in the boreal zone causes an increase in the annual runoff (Wei et al., 2022) but some studies reported also a decrease (e.g., Hou et al., 2022). Overall, the link between boreal forest harvesting and hydrological response is complex, and snow hydrology likely plays a large or determining role in the change of post-harvesting water balance (Wei et al., 2022).

Deeper discussion on the hydrological cycle doesn't really fit in this section of the introduction, but some points were added to the paragraph already dedicated for hydrological effects.

References:

Goeking, S. A. and D. G. Tarboton (2020). "Forests and Water Yield: A Synthesis of Disturbance Effects on Streamflow and Snowpack in Western Coniferous Forests." Journal of Forestry **118**(2): 172-192.

Wei, X., et al. (2022). "Forest harvesting and hydrology in boreal Forests: Under an increased and cumulative disturbance context." Forest Ecology and Management **522**: 120468.

Hou, Y., et al. (2022). "Cumulative forest disturbances decrease runoff in two boreal forested watersheds of the northern interior of British Columbia, Canada." Journal of Hydrology **605**: 127362.

Wei, X., et al. (2018). "Vegetation cover—another dominant factor in determining global water resources in forested regions." Global Change Biology **24**(2): 786-795.

**R1.4**        L46-47: please give a few examples of positive and negative feedbacks associated with forest-atmosphere exchange processes

See response to R1.2.

**R1.5**        L50-52: Estimates of the potential changes in C sink would be useful for context

See response to R1.2.

**R1.6**        L54-57: Estimated range of changes in surface T as a result of changes in albedo?

See response to R1.2.

**R1.7**        L59: How "significant" a source - please quantify and put in context

This sentence was setting up for the following sentence that provided numbers on the aerosol loading boreal forests can maintain. We've made some edits and additions to the text to put this more in context.

**R1.8**        L63: What is the estimated emission rate of BVOCs, and monoterpenes in particular, from boreal ecosystems and how does this compare with total global emissions?

Additions made to manuscript.

**R1.9**      L64-65: What does "comparatively potent" mean? Please give some values (atmospheric lifetimes, secondary organic aerosol formation potential, etc) for major monoterpene species.

Additions made to manuscript.

**R1.10**      L68: Can you give a relative increase in emissions for (say) a 2T or a 5TC rise in surface temperature?

The next comment is also related to this, and we have added estimates in connection with lines 84-89.

**R1.11**      L84-89: Can the authors give the current best estimate for increases in boreal biomass and BVOC emissions under future climate scenarios?

Some estimates were added to the text.

**R1.12**      L112-114: While I appreciate there is considerable uncertainty, it would still be good for the authors to give the range of possible values of cloud fractional cover between forested and open ground and between the seasons.

If we focus on boreal forest only, Duveiller et al. (2021), show a relative increase in cloud fractional cover of low-level convective clouds by ca 5% as compared to non-forested nearby areas during boreal summer and a decrease of similar magnitude during boreal winter. Xu et al. (2022) also observed cloud cover fraction decrease after forest loss in East Siberia during boreal summer by ca 0.2% (over years 2002-2018).

We also added estimates to the manuscript.

Xu R, Li Y, Teuling AJ, et al. Contrasting impacts of forests on cloud cover based on satellite observations. *Nat Commun*. 2022;13(1):670. Published 2022 Feb 3. doi:10.1038/s41467-022-28161-7

**R1.13**      L119: Specifically which properties will the authors focus on?

Sentence has been edited for specificity.

**Methods**:
**R1.14**      L131: It would be useful if the authors could mark this site on the biome map in Fig.1

The site was shown in the biome map but we had forgotten to mention it in the figure caption. It has now been updated.

**R1.15**      L135-137: Would it not have been of interest to see how properties of air masses with the same origin, and hence residence time over the forest, differed due solely to presence or lack of biogenic activity?

This would definitely be an interesting concept to study. We do not see how this is possible to execute in this environment, however, as we cannot simply "turn" the biogenic activity on or off. The biogenic activity is low in winter, but the meteorological conditions (+pollution) are so vastly different then that we do not see that comparing these two cases for example would be very informative for our purposes.

**R1.16**      L144-145: Why were these temporal and spatial resolutions used?

We wanted to use a relatively low arrival height, but not too low, where trajectories could "hit the ground" on their way and be cut short. 96-hour long trajectories were long enough for our purposes, and similarly the 1 h resolution for the air mass path locations.
Meteorological data output frequency used for the HYSPLIT model is a property of the archive model data (see e.g., Stein et al., 2015).
We utilised trajectories that had been produced already prior to this analysis, as we saw no pressing reason to produce new trajectories. The meteorological models used for HYSPLIT were selected from a few publicly available options in the NCEP archives that can be found through the website indicated in table 1. The first FNL archive (of (also) NCEP/GDAS output) ended in 2006, after which the change was made to GDAS one-degree data. In 2014 this was changed to the higher resolution 0.5-degree data. These are all commonly used data sets, and we find them all suitable for this analysis.

Edits to text: edit of language to make it clearer that the trajectories had been modelled before this project. Also "GFS data" was changed to "GDAS half-degree", as this is what it is called in the HYSPLIT data archive (although it also utilises GFS, which is why this abbreviation was previously used).

**R1.17**      L157: How large might the variation in error between trajectories be?

An estimate of 15-30% with a reference has been added here.

**R1.18**      L159-161: How many trajectories were analysed in total over the 11-year period? Given that this is already being reduced by the selection criteria, i.e. that it must lie 90% or more within the NW quadrant, I would suggest that the authors may need to carry out an uncertainty assessment to ensure the sample size is sufficiently large to reliably draw conclusions.

There were over 12 thousand trajectories (sum of monthly numbers shown in figure 2) in total that filled the selection criterion of coming from the north-west. We expect this to be high enough for drawing conclusions. The number can be limited much more by the number of measurements of certain variables that coincide with these trajectory arrival times. This is especially an issue with satellite data (discussed therein).
Generally, all the figures also show the number of observations in each bin to help the reader to infer which bin groupings are possibly more reliable than others.

**R1.19**      L170 (Fig. 1): Appears to show large tracts of "grassland" in the NW sector - is this the case? If so, it suggests that ToL is substantially over-estimating the time air masses spend above forest biomes, and that this "error" would be heavily dependent on precise air mass direction of travel.

The map has now been updated (Fig. 1 in the manuscript). Yes, there are also tracts of land with no forest cover, in the northern tundra and in the Scandes mountain range, but we do not think this presents a major issue for the interpretation that ToL can **approximately** describe the interaction time with the forest environment, as (1) the unforested area is still relatively narrow and (2) typical trajectory paths travel across rather than along them.

The 'Time-over-land' itself is essentially an approximation, based on a simplified (any single trajectory always is) travel path of incoming air and it is not, nor is it claimed to be, **exactly** the 'time of interaction with the forest environment', but just a simple easy-to-understand proxy for it, as forest is dominant in the area (Fig. 1).

The text in the manuscript was edited so that it matches the new updated figure.

We also acknowledged the tundra and mountain regions, which we argue to not be an issue, as they are also still vegetated and pristine, and importantly, quite narrow areas. We also explain again how ToL can be considered to describe interaction with a forest environment, because the forest coverage in the area is so high.

**R1.20**  L179: Why use median values rather than mean?

These meteorological quantities should not have very significant variation within the hour we are averaging over, so the difference between mean and median should be very small. However, generally, medians better capture the middle point (which we want) of the data, especially in the event that there would be outliers/unreliable data points or just an unsymmetrical distribution.

**R1.21**  L183-186: It is not clear where these instruments were located. Were they also deployed on the tower at the SMEAR II station?

The table has been updated with relevant information (measurement altitudes added).

**R1.22**  L212:How minor was this fraction? It seems that the authors have now listed such a multitude of reasons for data to be discarded that there must have been periods with very little useable data remaining in the dataset. It would be instructive to know how many observations / trajectories were analysed for each year, origin, etc and what fraction of the possible maximum number of observations this represents.

Data is only discarded for 3 reasons: 1) air mass is not in the clean sector; 2) The satellite image is limited (either the full image or the number of cloud pixels). Images can be partial also if the file cuts in the middle of our area of interest, but this happens only for 0.7% of all images; 3) observations are made at night (which can happen around midsummer in the area). This accounts only for 3.7% of observations, so we leave them out to focus on the more comparable daytime cases. The night-time observations are likely to have high uncertainties since the sun is low; at the same time, cloud properties are determined from the shortwave radiation.

Text was edited: "Files were processed file-by-file, and partial views resulting from subsequent files cutting over our area of interest (0.7% of all observations) were not combined into a single image. A few sporadic observations outside the main daytime time window (3.7% of all images) were also discarded both for consistency and because night-time observations are likely to be less reliable for measurements based on shortwave radiation."

The part about discarding cut files was deleted, as some of these partial/cut images, still have good enough coverage that one or both halves can be included.
Since the fraction of these cut files is so small, we do not see a reason for a more detailed analysis.

Some additional edits to the text for clarity were also made, and a sentence added to describe how, despite having several satellite images per day, we end up with a limited number of usable observations, because we are only selecting images with good enough spatial coverage that also need to match with a clean-sector trajectory.

To visualise the selection of data, the figure below shows the number of daytime satellite images with any number of pixels within our grid (green line) (cases with >5 or <4 observations only occur less than 1% of the time), number of observations averaged to even hours (blue). The markers show the number that coincide with a clean trajectory. Black markers show the number with also the required pixel coverage. Red markers are visible when there are more images that don't satisfy the criterion (otherwise the black markers are plotted

on top of them). Ultimately, we do not have very many observations left for analysis (only the number indicated by the black markers).

[Figure]

**R1.23** L217-219: How were the bins selected for each of the variables? Were the bins of equal length, equal number of observations, categorised by some other means (e.g. for temperature whether it was cold, average, warm or heatwave)?

The value ranges were based on percentiles (for most cases, not for precipitation as it would not be sensible because most of the cases have no precipitation). The text discussing these figures has now been edited so that this is easy for the reader to remember.
Groups in **Figure 5c** for temperature deviation, were also initially an exception, and were manually/arbitrarily selected, to make the midpoint between the second and third group zero, so that two groups showed only positive, and two only negative deviation. This was now modified, and data is similarly grouped based on quartiles. It is now thus consistent with the other figures.
 The width of the ToL bins, on the other hand, was based on data availability. For variables (such as the basic meteorological variables) that were available for nearly every trajectory, 1-hour ToL bins would still have a good number of observations, whereas for variables with much limited availability, 5-hour ToL bins are much more sensible. This can be easily seen from the "observations in bin" shown with the white line in each figure.

**R1.24** L254-262: On what basis have the authors selected the number concentrations for each of the 4 bins? Why choose 1400, 2200, etc? (I think this might be included in the caption for Fig 3 but should also be in the text)

The wording was edited for clarity and we also added already in the methods, a sentence describing how the grouping is based on quartiles, so that each group contains a quarter of the observations.
We also added similar information in connection with the manuscript rows
This information was also added to the indicated section of the text and language was modified to support this. See answer to comment R2.7.

**R1.25** L272-274: On what have the authors based their assertion that the fractions become relatively stable. It is not apparent from Fig 3 why not e.g. 50, 40, … Have the authors carried out rigorous statistical analysis to test this?

This was visually estimated, but now we added some data analysis to confirm. We agree that although the decrease in the fraction of observations in the bigger groups slows down at around 50 hours, it does not actually seem to fully stabilise. This is supported by the figure below, that shows the slope of an ordinary least-squares linear regression, fitted to $N_{CN}$ vs ToL data, onwards from the ToL starting point indicated by the x-axis. The circles show where the slope has a p value below 0.05.

This shows that even for $N_{CN}$ observations that have ToL>67.5 h, the slope of the regression fit is still negative in statistically significant (p<0.05) level. Only after this the fit is not statistically significant indicating that N_CN does not depend on ToL after that point.

We fitted a Pettitt's test to the bin medians, which indicated a change-point in the 50 h ToL bin. This is also the start point from which the gentlest slope can be fitted (marked with an 'x' in the figure below) (i.e. slope fitting the data with 42.5-95.5 h ToL (corresponding to bin edges)).

This slope is much gentler, than the slope fitting the observations in the 20*-45 h ToL bins, but still 24% of it.

(* We ignore the bins at the shortest ToLs that have an opposite trend)

We have now added explanations and descriptions of these tests to the manuscript. Now these numbers support (and finetune) what we could already interpreted visually from the figures. Specifically in this section, relating to N_CN, the text is edited to reflect this finding that a modest negative trend carries through the whole data, and we don't reach a static state. We however can find a change point at 50 h ToL with Pettitt's test, and from this point onward the trend in the data is much weaker than between air masses with ToL shorter than this.

[Figure]

**R1.26** L285-288: Again, it is not immediately apparent from Fig. 4 why the authors should select 20 and 60 hours as the transition times - it appears it would be equally valid to select 25 and 75. How were the two limiting times determined? By eye or statistically?

This was interpreted by visual estimation. The text was admittedly slightly confusingly organised. 20 hours is just the time when highest concentrations over-all are seen (indicated by the brightest/darkest red, which is in line with what we can see in Figure 3). Here we still

focus more on size than number, as the variation in numbers could already be seen in Figure 3.

The comparison between peak diameters was actually between 10 h and 60 h ToL bins (i.e. the time when concentration peak shifts to a bigger diameter, which can be easily seen).

After closer data-based inspection, we realised that it is more accurate to consider 55 h as the time after which the concentration peak remains relatively consistent, because after this point, the peak diameters vary only within 4.3% of this value.

The text has now been edited to reflect the fact that we can locate the "balancing" to the 55 h ToL bin. We also edited this section for clarity as it was written and structured in a confusing manner, and we link the findings of figure 3 and figure 4 more clearly.

**R1.27**     L302-303: Is this how the value of 60 hours was selected? That 95% of air masses have Nccn > lowest values? Still not clear from either the text or the figure (Fig 5 now).

The 60-hour mark was fully based on visual estimation, as it is roughly the time by which we can in most figures see a "balancing" of the fractions. But we have now added some data-based analysis to support this visual estimation. This included identifying a change point in the bin medians with Pettitt's test, and locating a start point for a flat least-square linear regression line:

For example, below is a scatter-plot for the $N_{CCN}$ (at S=0.2%) versus ToL. We fitted regression lines to the data from each ToL bin edge until the end (the longest ToLs), until we found the bin where we encounter a flat line (the slope figure on the right, where 'x' marks the bin edge). The figure on the right shows the slope (similarly to figure in response to R1.25). For the flat line, the variation (least-squares) is balanced on both sides, and there is "no relationship" any longer (which is why the p-value is also large). Therefore, we can say that after this point a further increase in ToL does not matter any longer, as the variable has already reached a balanced state with the environment.

(Here, we also added a reminder to the discussion of CCN figures in the manuscript that CCN at 0.2% is typically more relevant for real cloud formation.)

The Pettitt's test, which we fit to the bin medians (ignoring the shortest ToL bins) typically places the mean point in the earlier bin, but overall these two methods match quite well.

We have now edited the discussion relating to all of the figures where the balancing/static state were discussed, so that now we also have these two data-based estimations and their interpretation to support the visual estimation of the balancing of the fractions with respect to ToL.

The testing did fine tune the start point of this balancing, and now different variables have more precise estimates on the time after which they appear to reach a balanced state with the environment. In many cases this occurred already around 50 hours of ToL.

The manuscript with these new estimates has now been edited accordingly (including methods, results and conclusions).

[Figure]

**R1.28** L318-329: Same as before - how have the authors selected the values of q, T and RH to use as the end points of the bins? How have they ensured self-consistency between binning of these related variables?

With additions made to the text already in response to comment R1.24, and small edits to the figure caption: "The general description of this figure is the same as for **Error! Reference source not found.**, except that here the ToLs are binned into hourly bins, and group value limits in panel (c) are not based on percentiles but…", we think this should now be fairly clear to the reader. In this context, we do not fully understand what the reviewer means with self-consistency. The grouping is based on percentiles, and we investigate each variables connection to ToL individually.
This text discussing these figures has also been edited as discussed in previous response.

**R1.29** L335: Yes, although Figure 2 also showed that August had the greatest variability in ToL.

This sentence was edited for clarity, and we also acknowledged/mentioned the highest variability.

**R1.30** L366-367: Do the authors mean, the number of trajectories varies on any given day? Or the number of satellite overpasses? If the latter surely the authors could rank satellite products by coverage, reliability, etc and work hierarchically through them, i.e. ensuring there is not a time with >1 retrieval? Without this, surely they place too much emphasis on the meteorological and cloud conditions on those days in comparison to other cloudy days.

Some small edits were made to text for clarity.
The number of trajectories varies as we are only looking at clean-sector trajectories. Number of satellite overpasses (polar orbit) also slightly varies with the path of the satellite over the globe which is not exactly the same every day, and hence some days can have more images where the span of the satellite view extends over our small region.
Since we are looking at clean-sector trajectories only, this is of course a bigger reason for not having observations on many of the days. This happens if the day in general doesn't see clean-sector trajectories, or that they happen to come in at times when we do not have a coinciding satellite image.
There are occasionally several images from the same hour since we are looking at data from two separate satellite. This can also happen if the satellite image happens to be cut right over our location and if both of these "half-images" still meet the set criterion for coverage. In this

case we're taking a pixel-weighted average (see table), so that the "fuller" image is emphasised.

These were also discussed in response to comment R1.22

**R1.31** L390-392: Can the authors suggest how this might be done? Are they suggesting a similar approach but with a more rigorous or e.g. machine learning-based scrutiny of cloud retrievals? Or are they calling for ground-based observations of cloud fraction and thickness? The latter might, in particular, be a useful approach for discounting frontal clouds.

We are suggesting the addition of less data limited ground-based measurements, and have now also added this to the manuscript.

**R1.32** Fig. 9: It's not apparent that there is a break in the y-axis of either panel

True, figures without the break had accidentally slipped into the manuscript. Figures were meant to look like this, and have now been swapped. Now the y-axis has a break.

[Figure]

**R1.33** Fig. 9: What do the authors mean by "predefined"? Based on what criteria? And why was this not similarly done for all binning?

Dividing the precipitation data according to quartiles, like was done with the rest of the data, would not be useful here, as such a vast majority of cases have no precipitation at all (median is close to 0). Therefore, the group limits here were manually selected. They roughly correspond to what would be intensities of light drizzle, light rain, and higher rates, had the precipitation rate been the same for the whole time period.
The caption has now been updated, this was very confusingly phrased.

**R1.34** L468-470: Again, in-situ observations would help to clarify why there appears to be a break-down in the relationship at very high specific humidity.

Analysis shows no obvious explanation for this. (Fig 1.: RH is not consistently high in these

bins. Fig 2.: P (hPa), T (℃) and q: mostly higher T and P seen with these observations, which is not unexpected with high q and lack of precip. It is very apparent that the number of data points is small, and hence it is not surprising we might not have enough precipitating cases).

These bins have relatively low number of observations (and represent only 2% of all observations, all happening in only 47 days), bins 9.5 g/kg onwards each respectively having 70, 54, 28, 17, 7, 8, 3 and 1 observations.

We expect that precipitation, and especially high precipitation rates (that skew the mean more) are just too rare, and these low values are just down to random variation.

The following was deleted from the text as we have now checked that it is not true (RH in this bins is not exceptionally high which could suggest fog): "" And instead, we now say here that the precipitation is probably too rare, especially heavy rain, so that these bins with low number of observations do not have enough of them for reasonable conclusions, and the means are probably low due to random variation. Even the 9.5 g/kg bin has only 70 measurements, of which only 17 have any precipitation.

The line in the figure shows the number of observations in each bin, so that the reader can estimate the reliability of the averages. This is now also highlighted more clearly in the text.

[Figure]

**R1.35**    Fig. 11: While (a) does appear to support the hypothesis that higher specific humidity triggers high precipitation in the following hour, it would be useful to know why that only appears to be the case up to a specific humidity of 9g/kg. There seems no obvious reason why mean 1-hour precipitation should fall above this value. It could equally be the case that the 3 points between 8 and 9g/kg are outliers …

The points here are bin averages and the specific humidity bins between 8g/kg and 9g/kg are averaged over about 150-300 observations, so we do not think these can really be seen as "outliers" as such, but of course, they can still be less reliable than the earlier bins with much more data points, which is indicated by the line showing the number of observations.

Note however, that the number of observations approximately halves between 9 and 9.5 g/kg bins and when we go beyond 10g/kg, the number of observation gets so low that these are expected to be very unreliable (but are still shown for full transparency).

**R1.36**  Fig. 11: By contrast, panel (b) does not appear to show any robust trend whatsoever. Why have the authors included this in their analysis?

Panel (b) shows that there is a weak but statistically significant correlation between bin median precipitation and specific humidity. This relationship can be expected as an increase in specific humidity suggests more intensive rains. At the same time, we do not expect a strong relationship between these variables, since there are much more important drivers for precipitation than just local humidity.

**Conclusions**:
**R1.37**  L507-509: Presumably this is only the case for the very specific case of air masses travelling from the NW sector over Fennoscandia to arrive at Hyytiälä rather than being true for all locations?

We expect this to be applicable to other similar locations in the northern latitudes when airmasses travel from marine to forested areas. Naturally, we do not expect them to be applicable in very different climatic conditions or environments. In reality, air masses are of course often influenced by anthropogenic emissions as well, which is the case in Hyytiälä especially for air masses arriving from for example continental Europe, but here the focus was on the forests' influence specifically, and we expect at least the principles to be descriptive of the general phenomena.

**R1.38**  L512-515: What are the implications of forest interactions not having had time to take their full effect?

One thing is that we should be mindful that certain observations that are made in these types of areas shouldn't necessarily be generalised to wider regions. For example, if we measure some average aerosol values at SMEAR II, we shouldn't automatically expect them to reflect the whole boreal region and to be the same in the middle of Siberia, for example, when most of the clean air masses arriving to Hyytiälä have shorter ToLs that are required for a balanced state with the environment according to our analysis.

We have illustrated this point more in the text with an example, and hopefully it is clearer now what we are trying to say here.

**R1.39**  L520-522: It's really not clear how the authors' analysis and conclusions could be considered in forestry practices or plans for reforestation. How would they envisage policymakers and practitioners making use of the information and data presented here?

We agree that this was unclear and have now removed the sentence. Mainly, the point was that as this analysis provides information of forest atmosphere interactions, like for example the benefit of forests for inland water and effects on forest-aerosol-cloud interactions, which are part of the picture when thinking of total climatic effect of changes in forest cover. This could be beneficial for forest management planning.
But this sentiment is perhaps too loosely related to the investigation presented here, so we are just leaving it out.

**Additional changes:**

In addition to editing the conclusions section to reflect all the changes made to the manuscript we also edited the section so that our main findings are better articulated. Now, there is also hopefully a nicer linking to the topics we set up more clearly in the updated introduction. Some other minor edits have also been made elsewhere in manuscript, for clarity and to e.g. correct poor phrasing.

**Anonymous Referee #2**

The paper describes changes of aerosol properties, humidity and cloud properties that appear in originally clean marine air masses when they are transported over boreal forests in Scandinavia. The properties of the air masses under investigation stem from long-term observations with high quality instruments in Hyytiälä, Finland, as well as from MODIS satellite observations. The selection and differentiation of the large set of air masses is done based on back-trajectories calculated with the HYSPLIT model.

The study is of interest to the ACP readership and it is generally well presented. However, there are major questions about the findings related to water vapour and clouds and about the presented calculation of the "Time over Land (ToL)". Although this concept was used before in another publication (Petäjä, 2022), it needs to be clarified to what extent the given ToL really represents the time in which interactions between emissions from forests and the air mass finally analysed took place. A re-calculation of ToL may require a potentially time consuming new analysis of the data set. I recommend that the paper may be published after major revisions. This will also require that the authors either clarify my major concerns or that they re-evaluate their data set.

We would like to thank Referee#2 for their comments, our responses can be found below.

**Major comments:**

**R2.1**    Introduction: It remains unclear how the fact that the forest's properties are potentially changing with climate change is related to your study. This may of course be a motivation, but it is written in way that the reader expects new insights about changes in the air masses under investigation throughout the 11 years period that you look at.

We agree and have now deleted these sections from the text and edited the introduction to better match the result presented here. (See response to R1.2.)

**R2.2**    Line 96-98: I am irritated with this description of "recycled water". This is simply part of the global water cycle and both, evaporation from water surfaces like oceans and lakes, as well as evapotranspiration over land contribute to precipitation over continents.

We find it unfortunate if the reviewer is irritated by this, but as moisture recycling is a commonly used term (see e.g. Trenberth, 1999; Hornberger et al., 2014) and, first and foremost, the meaning of the word in this context is very easy to understand, so we think it is reasonable to use here. With the use of quotations around the word "recycled", we think that we already acknowledge to a reasonable degree, that this is not a perfect description, while also making up for it being used alone, instead of an established word pairing such as "moisture recycling" "precipitation recycling" etc.

Trenberth, K. E. (1999). "Atmospheric Moisture Recycling: Role of Advection and Local Evaporation." Journal of Climate **12**(5): 1368-1381.

Hornberger, G. M., Wiberg, P. L., Raffensperger, J. P., and D'Odorico, P.: Elements of physical hydrology, Second edi., Johns Hopkins University Press, Baltimore, Md, 2014.

**R2.3**    Line 138 – 165: You describe the "time over land" as the time a backward trajectory that arrives at Hyytiälä was at a geographical location that is somewhere over the land areas given in Figure 1. In addition, you assume that there will be some type of forest growing on this land surface and that this interacted with the air mass before it was characterised by the observations performed at (or over) Hyytiälä. Despite the fact that the

type of forest/trees that various air masses with similar ToL may have been in contact with will be very different, this interaction will only take place when the air mass did not travel in high altitudes. If the air mass travelled well above the planetary boundary layer, it is not likely that significant impacts of the underlying forest will be visible. Therefore, your ToL analysis may be better based on "time over land within the PBL" if you want to draw conclusion out of potential air mass modifications.

Firstly, we disagree with the statement that "the type of forest/trees [--] will be very different", as although, there are also narrow areas with mountain and tundra vegetation (see Fig. 1 in paper and response to R1.19), most of this land area is covered by boreal forest vegetation, which is dominated by evergreen needleleaf trees (with a small minority of deciduous trees). In this scale, any small internal variation (e.g. peatlands etc.) is acceptable being included as they are also a part of the larger boreal forest environment.

Secondly, yes, it is of course true, that air masses travelling at very high altitudes would not interact with the environment very much. We did perform an analysis where we calculated ToL only from points where the trajectory altitude was below the "mixing depth" output by the HYSPLIT model (which may not be a very accurate representation of the true PBL height, which can be quite complex to determine). As would be expected, the figures for these "ToL within (Hysplit) mixed layer" similarly show the processes earlier observed. This can be seen from the example below. (The "ToL within mixed" is naturally shorter than normal ToL, so there is a corresponding shift, note that the first and the last bins have extremely few observations.)

[Figure]

ToL in HYSPLIT mixed layer

We however, do not think that switching to this parameter would be beneficial. Firstly, as a concept, it can be more difficult to grasp, whereas the "time over land" is very simple. It is also consistent and directly **comparable with earlier publications** (e.g. Petäjä et al., 2022; Tunved et al., 2006; Liao et al., 2014), and it is **easy to transfer into a corresponding land area**. If one wanted to e.g. estimate the size of the "transitional zone" (marine-to-boreal-continental), we wouldn't even want to remove the internal variation in the air mass trajectory altitudes. We are also using such long datasets that we can expect these type of variations to somewhat "even out", so that we can end up with the average/typical conditions.
Also, any single trajectory is always highly simplified, as air masses do not truly travel in such a simple manner. Therefore, by default ToL for single trajectories is a crude approximation, which we again try to remedy by using a lot of data, so that the "noise" from single trajectory uncertainties is minimised as much as possible. Adding yet another uncertain parameter (the HYSPLIT trajectory mixing depth) would likely not be beneficial.

We also think that this criterion can be unnecessarily strict: While for daytime observations it could be relatively sensible, the night-time mixing layer is very thin. If an air mass is travelling in the residual layer, above the thin mixing layer, we can expect it to still be exposed to the

compounds emitted into the air from the surface during the day, even if it is at the time uncoupled with it (nighttime evapotranspiration and BVOC emittance are also lower in any case). (Although somewhat unrelated as they were looking at a very different scenario with high levels of atmospheric pollution, Hakala et al. (2022) applied a similar logic).

We do, however, agree with the reviewer that we should make sure that the trajectories do not travel so excessively high, that they would be most likely outside the daytime boundary layer. For this, we checked the fraction of trajectory ToLs that were also below different height levels. We found that on average (mean) 83% of trajectory points over land, were also below 500 metres (median: 91%), and the mean fraction below 1000 m was 95% (median 100%). Therefore, most of the trajectories travelled well below the typical daytime boundary layer heights within the growing season (e.g. Hyytiälä boundary layer analysis by Sinclair et al., (2022)). Therefore, we can assume that these air masses are indeed interacting with the surface.

We have now edited the text and added this information and discussion also to the manuscript.

Hakala, S., et al. (2022). "Observed coupling between air mass history, secondary growth of nucleation mode particles and aerosol pollution levels in Beijing." Environmental Science: Atmospheres **2**(2): 146-164.

Sinclair, V. A., Ritvanen, J., Urbancic, G., Statnaia, I., Batrak, Y., Moisseev, D., and Kurppa, M.: Boundary-layer height and surface stability at Hyytiälä, Finland, in ERA5 and observations, Atmos. Meas. Tech., 15, 3075–3103, https://doi.org/10.5194/amt-15-3075-2022, 2022.

**R2.4**      Line 168: Figure 1 does not demonstrate convincingly that there is enough forest existing in Fennoscandinavia to support your assumption that there is an influence of tree emissions on an air mass when the trajectory is over land.

We agree, we have now changed this map to European Forest Institute's Forest Map of Europe from 2011, that shows the forest cover percentage in high resolution.

**R2.5**      Line 195-211: As you explain, cloud fraction can be related to any type of cloud, not necessarily to low level clouds, only, for which a modification of their properties through aerosol particles observed at ground level can be expected. This might have a strong impact on your analysis that is not discussed. You might also try to get additional information on the height of the clouds and restrict your data set to low level clouds.

Yes, ideally, we would refine our analysis to cloud types we expect to be the most relevant. However, with the satellite data, we could only estimate the cloud base level indirectly from e.g. the cloud top height or cloud phase properties. The number of data pixels identified with our accepted cloud type in this manner would be limited by uncertainties associated with these methods (for example, phase identification can be left undetermined due to viewing anomalies, contamination or because of a multilayer cloud) (Platnick et al., 2018), or be unreliable due to assumptions that would have make. Unfortunately, the numbed of data points is also so low, that any refinement would unlikely yield any stronger conclusions. However, we do still expect that most clouds are low-level clouds (Ylivinkka et al. 2020).

These issues are brought up in the manuscript when discussing the results, and we do not see a reason to discuss them further already in the methods section. We only edited the text slightly, as the "cloud phase flag" sounds misleading, when cloud phases are reported by completely separate products (Platnick et al., 2018).

Ylivinkka, I., Kaupinmäki, S., Virman, M., Peltola, M., Taipale, D., Petäjä, T., Kerminen, V.-M., Kulmala, M., and Ezhova, E.: Clouds over Hyytiälä, Finland: an algorithm to classify clouds based on solar radiation and cloud base height measurements, Atmos. Meas. Tech., 13, 5595–5619, https://doi.org/10.5194/amt-13-5595-2020, 2020.

**R2.6** Line 195-211: Please give more details on the MODIS Level-2 Cloud Product. When is the overpass? Is this the "main time window" mentioned? What is the spatial resolution of individual cloud pixels? Some of the informations is given later but it should be presented here.

Edits have been made.

**R2.7** Line 254 (and in the following figures 3, 5, 6, 8, 9): On which basis do you define these 4 groups? Why don't you use median values and percentiles for each ToL bin?

The groups are based on percentiles, more specifically they are divided by quartiles (except for precipitation, which is dominated by zeroes), and the $2^{nd}$ quartile/50th percentile (separating the $2^{nd}$ smallest & $2^{nd}$ largest groups) corresponds then to the median value of the observations. This was stated in the caption of Figure 3., but now it has been added clearly to the text already in the methods. See responses to R1.23 and R1.24.
The text was also edited in many places to remind the reader that this is the grouping, so that it is easy to keep in mind. References to quartiles are now made for example so that instead of "nearly half of them have number concentrations below 1400 cm$^{-3}$" we say "nearly half of them have number concentrations below the lower quartile (below 1400 cm$^{-3}$)"). Similar editing was done in all the places where these types of figures are discussed.

**R2.8** Line 318-329: In my opinion, you cannot conclude that air masses with longer ToL have higher specific humidity (q) because you most likely only see a temperature effect, with warmer air masses having higher saturation pressures. You discuss this, but still I think that you should avoid giving the impression that there is a causal relationship between ToL and q. This is also supported by the fact that you do not see a clear relationship with RH. In the analysis of the temperature dependence on ToL you also mix effects of seasonality with weather patterns and you neglect effects of vertical air mass transport.

The writing in this section has been edited to hopefully make it clearer and easier to follow, as it was initially perhaps too confusing. Hopefully now the main message, interpretation and hypotheses we make, are more apparent.
We think that it is fair to conclude that the specific humidity is higher at longer ToL, as this is indeed what we see. As mentioned here, we also already discuss the fact that this is most likely facilitated by warming. Meaning, that the warming means that the air mass CAN uptake and hold more water vapour. This still means that we still also need a source of water vapour, so that the air mass also WILL have an increase in moisture content. This is where the water vapour emitted though forest evapotranspiration comes in. RH probably does not show a significant increase, because the evapotranspiration is strong enough to sustain similar RH levels in warmer air masses but is not so strong that it would clearly shift the RH distribution further.

We do not fully understand the reason behind the comment "you also mix effects of seasonality with weather patterns", since the point here is that there are seasonal differences in atmospheric flow (incl. weather patterns), which are exactly what we wish to balance with the comparison to the monthly mean. Unfortunately, we are not entirely clear on how the reviewer wished us to consider vertical air mass transport in this context.

(Figure 5c group limits were also edited to make it consistent with other figures, see answer to R1.23)

**R2.9**       Figure 6: It remains unclear why you reduce the bin size to 1 h.

Simply because the data availability is better, and this can be done. This was added to the caption.

**R2.10**      Figure 7: What can I learn from this figure? It just shows the Clausius Clapeyron equation and that relative humidities cover a wide range at a given temperature.

It is mostly a reminder and a visual aid, as q-T relationship is an important part of the discussion here, and we want to help the reader to effortlessly follow it. (Now we also call it a "reminder" in the text so that it is clear that this is the reason why we are showing it).

**R2.11**      Line 370: "A modest increase seems to occur after ToL exceeds approximately 50 hours.": You should check whether these differences are statistically significant.

We identified, that from the 50 h ToL bin, the median COT data could be divided into two groups for which a two-sample t-test (Matlab ttest2) on log-transformed distributions suggested a statistically significant ($p<0.05$) difference.
Below is a figure with 1) the distributions of COT observations in 20-45 h ToL bins and in 50-90 h ToL bins 2) the log-transformed distributions, 3) normal probability plot (for all datapoints, 20-90h bins) and 4) testing for the separation of the median COT into 2 groups from different points, where we found that when separating the observations into groups consisting of observations in 20-45 h ToL bins, and observations in the 50-90 h ToL bins, the difference between the two groups is statistically significant ($p<0.05$).

(We started from the observations in the 20 h ToL bin because we were mainly interested in these values, and not in the very shortest ToL values, which many of the figures show, can be typically quite different)

Overall, the limited cloud data does not allow us to draw strong conclusions (as discussed in the manuscript), which is why we must largely leave adding detail to these issues to possible future analyses, but the t-test at least offers some support for our visual estimations regarding this figure.

Whereas, for CF, with the limited data and complicated distribution, we do not think that we can confidently determine a robust statistical significance but conclude that more data is required.

Relevant edits were made to the manuscript.

[Figure]

**R2.12**    Section 3.3: It looks like your analysis of COT and CF does not reveal any dependence on ToL: This might be connected with the relatively low number of cases that you could analyse, but it may also be related to the fact that you mix all types of clouds.

We agree with the reviewer. The satellite-related analysis remains an open question, due to the limited amount of data. Separation between different types of clouds would make this data set even more limited. Therefore, we leave this issue for the future analysis preferably using ground-based observations of cloudiness (e.g., CLOUDNET data base).

**R2.13**    Section 3.4: Again, you should check whether the differences you see are statistically significant. In addition, one would expect that ToL mainly reflects different weather patterns and is not a variable that can explain the amount of rainfall observed at Hyytiälä.

For the 1 hour precipitation accumulation, Pettitt's test fitted to bin means gives a statistically significant change point (p>0.05), but not for 3-hour precipitation (p~0.16). Overall, a linear least-square regression fit to precipitation data points (>17.5h) produces a positive trend (naturally, a very small one) that is statistically significant.

Relevant edits were made to the manuscript.

And yes, weather patterns are also reflected in ToL, as it is not possible to "isolate" air mass transport from large scale atmospheric flow that also includes weather patterns. We also see (and address) this in the precipitation figures, and discuss the high precipitation frequencies at very short ToL, which are likely to be connected to frontal precipitation, where air masses are moving quickly.
We definitely would not expect a single trajectory ToL to be able to predict the precipitation probability associated with that single air mass, as there are much more significant factors (like weather patterns), but rather, by analysing large groups of air masses, we can identify the smaller underlying signals that connect the air mass properties to the surface interaction.

Air mass travel path, weather patterns and surface interactions are completely interconnected, and we cannot separate them in the real world. Therefore, features of weather patterns can indeed be reflected in the ToL, however, we do not think that this would negate the fact that the humidity for the increased precipitation rates at longer ToL, could originate in the evapotranspiration from the forest surface that these air masses spend longer times interacting with.

**R2.14** Line 442 – 450 and Figure 10: In my opinion, you again see mainly the dependence of water vapour saturation pressure (and therefore q, CWT and also precipitation rate) on temperature. It is misleading when you try to explain differences in arriving air masses with ToL.

These changes the air masses undergo as the travel over land and are due to concurring effects. We have already described previously that we expect the increase in specific humidity to be largely facilitated by the increasing temperatures enabling more efficient evapotranspiration. In this section, we merely show that this moisture appears to also translate to cloud water path in the whole column and eventually precipitation, "completing" this chain of events.

**R2.15** Line 473/474: This sentence is obscure and needs to be rephrased. Nevertheless, I think it correctly says what you see and the ToL analysis that you put on top of it does not reveal new insights or links with the aerosol size distribution that you want to demonstrate (according to the introduction).

Unfortunately, wrong row numbers have probably slipped into this comment, and we do not know which part of the text this comment is referring to.
However, we strongly edited the conclusions section to match all the changes made in response to the referee comments, and to also write the main conclusions and findings in a clearer manner, we hope that we also made changes to this specific part of the text that needed rephrasing.

**R2.16** Line 501 – 504: I cannot see where you showed a link between cloud reflectance and CCN in your study.

Indeed, this was poorly phrased. We only know from literature that the CCN we observe increasing with ToL could have the potential to make the clouds in these air masses more reflective. COT is also strongly connected to reflectance, and we observed some indications (although uncertain) that it could on average be higher in long ToL air masses as well.
The conclusions section has been edited significantly overall, to reflect all the changes made to the manuscript and to report our findings more clearly.

**Minor comments**
**R2.17** Line 43: the forest ✓
**R2.18** Line 60/61: "climatically relevant sizes" is a not well defined expression. Please rephrase. ✓
**R2.19** Line 65: which VOCS do you have in mind when you mention "other common BVOCs"? Please add examples. ✓
**R2.20** Line 72-74: "While not exclusive …". This sentence is unclear. Please explain in a different way what you want to say. ✓
**R2.21** Line 80: correct „potentiantially" ✓
**R2.22** Line 85: correct „photosynthesising"
   **Re:** It is already correct.

**R2.23** Line 237: I assume this is from 1 April to 30 September. You should exactly specify this. ✓

**R2.24**      Line 269: remove "s". ✓

**R2.25**      Line 286: rephrase "until a seemingly balanced is reached". ✓

**R2.26**      Line 305 appear to `be` ✓

**R2.27**      Line 366: "The number of successful daily observations also varies slightly, which may also lead to some days having better representation than others.": It remains unclear what you want to say.

     **Re:** This sentence was deleted. We agree that it was somewhat obscure, and considering that the number of satellite datapoints per day, only varies typically between 2-4, it is not very relevant, as what matters more for how well represented certain days are in the data, is whether the air masses are coming from the clean sector or not (which is already discussed in the text).

**R2.28**      Line 368: is `shown` ✓

**R2.29**      Line 470: omit the first "was" ✓

**R2.30**      Line 509: `established` ✓

We have also made the edits requested in the minor comments